# Variational Inference for Continuous-Time Switching Dynamical Systems

**Lukas Köhs**      **Bastian Alt**      **Heinz Koeppl**
Department of Electrical Engineering and Information Technology
Technische Universität Darmstadt
`{lukas.koehs, bastian.alt, heinz.koeppl}@bcs.tu-darmstadt.de`

## Abstract

Switching dynamical systems provide a powerful, interpretable modeling framework for inference in time-series data in, e.g., the natural sciences or engineering applications. Since many areas, such as biology or discrete-event systems, are naturally described in continuous time, we present a model based on a Markov jump process modulating a subordinated diffusion process. We provide the exact evolution equations for the prior and posterior marginal densities, the direct solutions of which are however computationally intractable. Therefore, we develop a new continuous-time variational inference algorithm, combining a Gaussian process approximation on the diffusion level with posterior inference for Markov jump processes. By minimizing the path-wise Kullback-Leibler divergence we obtain (i) Bayesian latent state estimates for arbitrary points on the real axis and (ii) point estimates of unknown system parameters, utilizing variational expectation maximization. We extensively evaluate our algorithm under the model assumption and for real-world examples.

## 1 Introduction

Many natural and engineered dynamical systems can be understood in terms of continuous-discrete hybrid models, in which a given system switches between discrete modes exhibiting continuous dynamics. Examples include neuro-mechanical models of locomotion [1], transition dynamics between different brain states [2, 3] and brain-state dependent decision-making in neuroscience [4]; regime-switching volatility dynamics [5] and risk assessment [6, 7] in financial analysis or electric power systems [8]; and phenotype differentiation in systems biology [9].

In a discrete-time setting, a widely used class of stochastic hybrid models are switching linear dynamical system (SLDS) [10], which have received considerable attention in recent years [11, 12, 13, 14, 15]. Since real-world physical and biological systems naturally evolve in continuous time, a discrete-time description of such systems is however limiting. In biological experiments and discrete-event systems in engineering [16], for instance, one typically (i) is interested in the system behavior at any given point in time and (ii) can not easily determine an appropriate time discretization. To overcome these limitations, a continuous-time analog to SLDS models has been put forward termed switching stochastic differential equations (SSDEs) [17], which augment a set of diffusion processes with an underlying Markov jump process (MJP). Hybrid models of this kind have a long tradition in statistics [18] and have been analyzed in particular in applications to biological systems [19].

Diffusion processes and MJPs have been treated extensively in the literature [20, 21]. For each process class individually, exact expressions for the posterior paths given some set of observations can be obtained [22]. However, for diffusion processes in particular, these expressions quickly become intractable as they entail solving multi-dimensional partial differential equations (PDEs). This is

35th Conference on Neural Information Processing Systems (NeurIPS 2021).

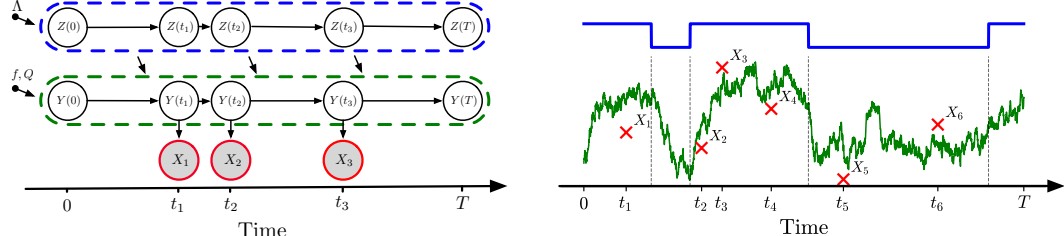

Figure 1: Sketch of the three layers of the hybrid process model. A two-state MJP $Z(t)$ (blue) evolves freely in the time interval $t \in [0, T]$, see Eq. (1) . The MJP controls the dynamics of the SSDE $Y(t) \mid Z(t)$ (green), see Eq. (2). From these continuous dynamics, only noisy observations $X_1, X_2, \ldots$ (red) are available for inference at irregularly-spaced time points $t_1, t_2, \ldots$. Left: Graphical model. Right: Sample path (vertical dashed lines indicate the $Z$-transitions).

aggravated if the diffusion is coupled to an underlying jump process, because both processes then have to be solved jointly.

One way to circumvent this issue are Monte Carlo approaches, which have been devised for both diffusion [23] and jump processes [24]. Sampling methods may however suffer from slow convergence, and, as we shall show, still face computational intractabilities for hybrid systems. An established approach avoiding these problems are variational inference (VI) methods, which approximate the exact posteriors by optimization [25]. For inference and parameter learning in diffusion processes, continuous-time VI frameworks have been developed utilizing, e.g., Gaussian processes (GPs) [26, 27], and general exponential family distributions [28]. Similar methods have also been devised for inference in MJPs [29, 30]. To the best of our knowledge, an inference framework for continuous-time hybrid processes is however lacking. We draw on these previous works and present a generalized VI framework for hybrid systems which recovers existing diffusion and MJP approximations as special cases. We specifically focus on meta-stable systems which remain in distinct, qualitatively different regimes over extended periods of time, which are of special interest, e.g., in computational structural biology [31]. An implementation of our proposed method is publicly available.[1]

## 2 Mathematical Background

### 2.1 The Model

In this work, we consider three joint stochastic processes $\{Z(t)\}_{t \geq 0}$, $\{Y(t)\}_{t \geq 0}$, and $\{X_i\}_{i \in \mathbb{N}}$. A continuous-time adaptation of a probabilistic graphical model and a realization of the processes is shown in Fig. 1.

**The Switching Process.**  The discrete-valued process $Z(t) \in \mathcal{Z} \subseteq \mathbb{N}$ is given as a latent Markov jump process (MJP) freely evolving in time $t$. An MJP is a continuous-time Markov process [32] on a countable state space $\mathcal{Z}$ and is completely characterized by an initial probability $p_0(z) := \mathbb{P}(Z(0) = z)$, $\forall z \in \mathcal{Z}$ and a transition rate function

$$\Lambda(z, z', t) = \lim_{h \searrow 0} \frac{\mathbb{P}(Z(t+h) = z' \mid Z(t) = z)}{h} \tag{1}$$

for $z' \in \mathcal{Z} \setminus z$, with the *exit rate* $\Lambda(z, t) := \sum_{z' \in \mathcal{Z} \setminus z} \Lambda(z, z', t)$.

**The Subordinated Diffusion Process.**  The freely evolving MJP controls the dynamics of a continuous-valued process $Y(t) \in \mathcal{Y} \subseteq \mathbb{R}^n$, which is given as a latent switching stochastic differential equation (SSDE) in an Itô sense, [17]

$$\mathrm{d}Y(t) = f(Y(t), Z(t), t)\,\mathrm{d}t + Q(Y(t), Z(t), t)\,\mathrm{d}W(t), \tag{2}$$

where $f : \mathcal{Y} \times \mathcal{Z} \times \mathbb{R}_{\geq 0} \to \mathcal{Y}$ is an arbitrary drift function, the dispersion $Q : \mathcal{Y} \times \mathcal{Z} \times \mathbb{R}_{\geq 0} \to \mathbb{R}^{n \times m}$ determines the noise characteristics of the system and $W(t) \in \mathbb{R}^m$ is a standard Brownian motion.

---

[1]https://git.rwth-aachen.de/bcs/projects/lk/vi-ct-shs.git

We define the system noise covariance $D(Y(t), Z(t), t)$ as $D := QQ^\top$. Note that the only difference to a conventional stochastic differential equation (SDE) is the $Z(t)$-dependence of the drift function and dispersion; hence, Eq. (2) can be understood as a collection of individual SDEs between which the systems switches via $Z(t)$. For an accessible introduction to SDEs, see [33].

**The Hybrid Process.** In the following, we refer to the continuous value $Y(t)$ as the *state* and to the discrete value $Z(t)$ as the *mode* of a hybrid process. The hybrid process $\{Z(t), Y(t)\}_{t \geq 0}$ is fully characterized by its time-point wise marginal density $p(y, z, t) := \partial_{y_1} \cdots \partial_{y_n} \mathbb{P}(Y(t) \leq y, Z(t) = z)$, where "$\leq$" has to be interpreted element-wise. This density is given as the solution to the hybrid master equation (HME) [34, 35]

$$\partial_t p(y, z, t) = \mathcal{A}p(y, z, t), \tag{3}$$

with initial condition $p(y, z, 0) = p_0(z)p_0(y, z)$, where $p_0(y, z) := \partial_{y_1} \cdots \partial_{y_n} \mathbb{P}(Y(0) \leq y \mid Z(0) = z)$ denotes the initial density of the $Y$-process and $p_0(z)$ the initial probability mass function of the $Z$-process. The operator $\mathcal{A}(\cdot) = \mathcal{F}(\cdot) + \mathcal{T}(\cdot)$ is given via

$$\mathcal{F}\phi(y, z, t) := -\sum_{i=1}^{n} \partial_{y_i} \{f_i(y, z, t)\phi(y, z, t)\} + \frac{1}{2}\sum_{i=1}^{n}\sum_{j=1}^{n} \partial_{y_i}\partial_{y_j}\{D_{ij}(y, z, t)\phi(y, z, t)\}$$

$$\mathcal{T}\phi(y, z, t) := \sum_{z' \in \mathcal{Z} \setminus z} \Lambda(z', z, t)\phi(y, z', t) - \Lambda(z, t)\phi(y, z, t),$$

for an arbitrary test function $\phi : \mathbb{R}^n \times \mathcal{Z} \times \mathbb{R}_{\geq 0} \to \mathbb{R}$. We provide a detailed derivation in Appendix A.1. As the discrete process $Z(t)$ is independent of $Y(t)$, we further obtain the dynamics of the marginal distribution $p(z, t) := \mathbb{P}(Z(t) = z)$ by integrating over the continuous variable $Y$ in the HME. This recovers the traditional master equation [32]

$$\frac{\mathrm{d}}{\mathrm{d}t}p(z, t) = \sum_{z' \in \mathcal{Z} \setminus z} \Lambda(z', z, t)p(z', t) - \Lambda(z, t)p(z, t), \tag{4}$$

with $p(z, 0) = p_0(z) \ \forall z \in \mathcal{Z}$; for details, see Appendix A.1.1.

Note that a general, analytical solution to the PDE (3) does not exist. Numerical solvers utilizing schemes such as the finite differences or finite element method suffer from the curse of dimensionality and can in principle only be applied to very low-dimensional state spaces [36] . Even in low dimensions however, solvers need to be adapted to the problem at hand and may struggle due to, e.g., slow step-size adaptation. On the other hand, sampling trajectories from a hybrid process $\{Y(t), Z(t)\}$ is straightforward: One can draw the process $Z(t)$ by utilizing the Doob-Gillespie algorithm [37, 38]. Given this trajectory, the SSDE $Y(t) \mid Z(t)$ can be simulated using, e.g., an Euler-Maruyama or stochastic Runge-Kutta method [39].

**The Observation Process.** Lastly, we denote with $\{X_i\}_{i \in \mathbb{N}}$ the countable set of observed data points at times $\{t_i\}_{i \in \mathbb{N}}$. The observations $X_i \in \mathcal{X}^l$ are generated as $X_i \sim p(x_i \mid y_i)$, where $p(x_i \mid y_i) := \partial_{x_{i1}} \cdots \partial_{x_{il}} \mathbb{P}(X_i \leq x_i \mid Y(t_i) = y_i), i \in \mathbb{N}$, by conditioning on the diffusion process $Y(t)$. The observation space $\mathcal{X}$ can be either discrete, $\mathcal{X} \subseteq \mathbb{N}$, or continuous, $\mathcal{X} \subseteq \mathbb{R}$. Note that in our model, a continuous-time description for the latent processes is assumed, while the observations are recorded at discrete time points. It therefore belongs to the class of continuous-discrete models, which have a long history in the filtering community [33, 40, 41]. This type of description is of great practical relevance as data is often recorded at discrete time points while the system of interest in fact evolves continuously in time, see, e.g., [16].

## 2.2 Exact State Inference

We now show how the exact posterior inference problem is solved in principle; the detailed derivations can be found in Appendix A.2.

The inference problem consists of finding the posterior hybrid process $\{Z(t), Y(t) \mid x_{[1,N]}\}$, where we condition on a finite set $x_{[1,N]} = \{x_1, \ldots, x_N\}$ of $N$ observations obtained at time points $\{t_1, \ldots, t_N\}$ in the interval $[0, T]$. The posterior process is fully specified by its marginal density

$p(y, z, t \mid x_{[1,N]}) := \partial_{y_1} \cdots \partial_{y_n} \mathbb{P}(Y(t) \leq y, Z(t) = z \mid X_1 = x_1, \ldots, X_N = x_N)$, which is known as the *smoothing distribution*. The smoothing distribution is given by

$$p(y, z, t \mid x_{[1,N]}) = C^{-1}(t)\alpha(y, z, t)\beta(y, z, t), \tag{5}$$

with the filtering density $\alpha(y, z, t) := \partial_{y_1} \cdots \partial_{y_n} \mathbb{P}(Y(t) \leq y, Z(t) = z \mid X_1 = x_1, \ldots, X_k = x_k)$, the backward density $\beta(y, z, t) := \prod_{i=1}^{l} \prod_{j=k+1}^{N} \partial_{x_{j_i}} \mathbb{P}(X_{k+1} \leq x_{k+1}, \ldots, X_N \leq x_N \mid Y(t) = y, Z(t) = z)$ and a time-dependent normalizer $C(t) = \sum_z \int \alpha(y, z, t)\beta(y, z, t)\, \mathrm{d}y$, where $k = \max(k' \in \mathbb{N} \mid t_{k'} \leq t)$. The components $\alpha$, $\beta$ and $C$ are continuous-time analogs to the quantities of the forward-backward algorithm for discrete-time hidden Markov models (HMMs) [10].

It is a standard result for continuous-discrete filtering problems [33] that the filtering distribution between observation time points follows the prior dynamics, $\partial_t \alpha(y, z, t) = \mathcal{A}\alpha(y, z, t)$, with initial condition $\alpha(y, z, 0) = p_0(z)p_0(y, z)$. At the observation times, it is reset as $\alpha(y, z, t_i) = \tilde{C}_i^{-1}\alpha(y, z, t_i^-)p(x_i \mid y)$, with the normalizer $\tilde{C}_i = \sum_z \int \alpha(y, z, t_i^-)p(x_i \mid y)\, \mathrm{d}y$ and $\alpha(y, z, t_i^-) := \lim_{h \searrow 0} \alpha(y, z, t_i - h)$. Similarly, the backward distribution between observations is given as the solution to another PDE [22]

$$\partial_t \beta(y, z, t) = -\mathcal{A}^\dagger \beta(y, z, t), \tag{6}$$

with end point condition $\beta(y, z, T) = 1$ and adjoint operator $\mathcal{A}^\dagger$, see Appendix A.1. The reset conditions at observation times are given as $\beta(y, z, t_i^-) = \beta(y, z, t_i)p(x_i \mid y)$.

By calculating the time derivative of Eq. (5), it can be shown (see Appendix A.2.3) that the smoothing distribution itself follows a HME

$$\partial_t p(y, z, t \mid x_{[1,N]}) = \tilde{\mathcal{A}} p(y, z, t \mid x_{[1,N]}), \tag{7}$$

with initial condition $p(y, z, 0 \mid x_{[1,N]}) \propto p_0(z)p_0(y, z)\beta(y, z, 0)$. The operator $\tilde{\mathcal{A}}$ contains the posterior drift function $\tilde{f}_i(y, z, t) = f_i(y, z, t) + \sum_{j=1}^{n} D_{ij}(y, z, t)\partial_{y_j}\{\log \beta(y, z, t)\}$, the dispersion matrix $\tilde{D}(y, z, t) = D(y, z, t)$ and the posterior rate function $\tilde{\Lambda}(z', z, t) = \Lambda(z', z, t)\frac{\beta(y, z, t)}{\beta(y, z', t)}$.

## 3 Approximate Inference

Since the smoothing distribution is governed by the HME (7), which depends on the solution of Eq. (6), the exact inference problem amounts to solving two PDEs, which is computationally intractable already for toy systems. Similarly, a naïve posterior sampling scheme would still require solving the backward PDE Eq. (6) and hence suffers from the same issue [42]. To address this challenge, we adopt a VI approach: we aim to find an approximate path measure $\mathbb{Q}_{Y,Z}$ that minimizes the path-wise Kullback-Leibler (KL) divergence

$$\mathrm{KL}\left(\mathbb{Q}_{Y,Z} \mid\mid \mathbb{P}_{Y,Z\mid X}\right) = \mathsf{E}_{\mathbb{Q}_{Y,Z}}\left[\log \frac{\mathrm{d}\mathbb{Q}_{Y,Z}}{\mathrm{d}\mathbb{P}_{Y,Z\mid X}}\right], \tag{8}$$

where $\frac{\mathrm{d}\mathbb{Q}_{Y,Z}}{\mathrm{d}\mathbb{P}_{Y,Z\mid X}}$ is the Radon-Nikodym derivative between $\mathbb{Q}_{Y,Z}$ and the exact posterior measure $\mathbb{P}_{Y,Z\mid X}$ over paths $Y_{[0,T]} := \{Y(t)\}_{t \in [0,T]}$ and $Z_{[0,T]} := \{Z(t)\}_{t \in [0,T]}$. For details on the path-wise KL divergence between stochastic processes, see, e.g., [28, 43, 44]. It is a standard result for VI methods that Eq. (8) can be recast as [25, 42]

$$\mathrm{KL}\left(\mathbb{Q}_{Y,Z} \mid\mid \mathbb{P}_{Y,Z\mid X}\right) = \mathrm{KL}\left(\mathbb{Q}_{Y,Z} \mid\mid \mathbb{P}_{Y,Z}\right) - \mathsf{E}_{\mathbb{Q}_{Y,Z}}[\ln p(x_{[1,N]} \mid y_{[0,T]})] + \log p(x_{[1,N]}), \tag{9}$$

with the expected log-likelihood $\mathsf{E}_{\mathbb{Q}_{Y,Z}}[\ln p(x_{[1,N]} \mid y_{[0,T]})] = \sum_{i=1}^{N} \mathsf{E}_{\mathbb{Q}_{Y,Z}}[\ln p(x_i \mid y_i)]$. The minimization problem over Eq. (8) can then be cast as a maximization problem over the evidence lower bound (ELBO) [25]

$$\mathcal{L}[\mathbb{Q}_{Y,Z}] = \mathsf{E}_{\mathbb{Q}_{Y,Z}}[\ln p(x_{[1,N]} \mid y_{[0,T]})] - \mathrm{KL}\left(\mathbb{Q}_{Y,Z} \mid\mid \mathbb{P}_{Y,Z}\right), \tag{10}$$

which does not include the computationally intractable marginal log-likelihood $\log p(x_{[1,N]})$ and can hence be evaluated.

Optimizing Eq. (10) requires explicitly computing the path-wise KL divergence $\mathrm{KL}\left(\mathbb{Q}_{Y,Z} \mid\mid \mathbb{P}_{Y,Z}\right)$. For two hybrid processes of the same class obeying Eq. (3), this expression can formally be derived

using Girsanov's theorem for diffusion processes [45] and MJPs [46]. A more intuitive derivation can however be carried out using a limiting argument, similar to [29]. The detailed derivation can be found in Appendix A.3.1. Assuming a constant, state- and mode-independent dispersion $D$ as done also in [27] and a drift $g(y, z, t)$ and rate function $\tilde{\Lambda}(z, z', t)$ pertaining to the variational measure $\mathbb{Q}_{Y,Z}$, the path-wise KL divergence is obtained as

$$
\text{KL}\left(\mathbb{Q}_{Y,Z} \,\|\, \mathbb{P}_{Y,Z}\right) = \text{KL}\left(\mathbb{Q}_{Y,Z}^0 \,\|\, \mathbb{P}_{Y,Z}^0\right) + \frac{1}{2}\int_0^T \mathsf{E}\left[\|(g(y,z,t) - f(y,z,t)\|_{D^{-1}}^2\right.
$$

$$
\left. + \sum_{z' \in \mathcal{Z}\setminus z} \left\{\tilde{\Lambda}(z,z',t)\left(\ln\tilde{\Lambda}(z,z',t) - \ln\Lambda(z,z',t)\right)\right\} - (\tilde{\Lambda}(z,t) - \Lambda(z,t))\right] \mathrm{d}t, \qquad (11)
$$

with the weighted norm $\|x\|_A^2 := x^\top A x$, the KL of the initial distributions, $\text{KL}\left(\mathbb{Q}_{Y,Z}^0 \,\|\, \mathbb{P}_{Y,Z}^0\right)$, and the expectation is carried out with respect to the variational time-point marginal $q(y, z, t) := \partial_{y_1}\cdots\partial_{y_n}\mathbb{Q}(Y(t) \leq y, Z(t) = z)$. We note that extensions to mode- and time-dependent $D = D(z, t)$ [45] or state-dependent $D = D(y)$ [28] are also possible. Additionally, in the absence of any coupling between $Z(t)$ and $Y(t)$, i.e., $f(y, z, t) = f(y, t)$, Eq. (11) reduces to the sum of the known individual path-wise KL divergences for diffusion processes and MJPs.

### 3.1 The Constrained Objective

Since Eq. (10) is a mere reformulation, it is still optimized by the true, intractable posterior distribution $\mathbb{P}_{Y,Z|X}$. To arrive at computationally tractable expressions, we restrict the class $\mathcal{Q}$ of admissible variational processes. Making a structured mean-field ansatz, we approximate the exact joint posterior density $p(y, z, t \mid x_{[1,N]})$ as

$$
\begin{aligned}
p(y, z, t \mid x_{[1,N]}) &= p(z, t \mid x_{[1,N]}) \cdot p(y, t \mid z, t, x_{[1,N]}) \\
&\approx q_Z(z, t) \cdot q_Y(y, t \mid z) =: q(y, z, t),
\end{aligned} \qquad (12)
$$

with $p(y, t \mid z, t, x_{[1,N]}) := \partial_{y_1}\cdots\partial_{y_n}\mathbb{P}(Y(t) \leq y \mid Z(t) = z, x_{[1,N]})$. We approximate the exact conditional $p(y, t \mid z, t, x_{[1,N]})$, which in general does not have a simple parametric form, by one fixed parametric expression per mode $q_Y(y, t \mid z) := \partial_{y_1}\cdots\partial_{y_n}\mathbb{Q}(Y(t) \leq y \mid \tilde{Z} = z)$ via the introduction of the time-independent random variable $\tilde{Z}$. This results in a point-wise mixture distribution $q(y, z, t)$ with weights $q_Z(z, t) := \mathbb{Q}(Z(t) = z)$ and mixture densities $q_Y(y, t \mid z)$. Note that this is similar in spirit to amortized inference techniques [47], because we utilize the same parametric form for all times $t$.

To ensure the mixture distribution structure Eq. (12) to hold at every time point $t$, we impose separate constraints on the dynamics of $q_Z(z, t)$ and $q_Y(y, t \mid z)$. Firstly, we require the marginal $q_Z(z, t)$ to obey a master equation

$$
\frac{\mathrm{d}}{\mathrm{d}t}q_Z(z, t) = \sum_{z' \in \mathcal{Z}\setminus z} \tilde{\Lambda}(z', z, t)q_Z(z', t) - \tilde{\Lambda}(z, t)q_Z(z, t), \ \forall z \in \mathcal{Z}. \qquad (13)
$$

This reproduces the structure of the exact posterior marginal $p(z, t \mid x_{[1,N]})$, which can be seen by integrating out the continuous variable in the HME (7), c.f. Eq. (4). Secondly, we constrain the variational factor $q_Y(y, t \mid z)$ to follow a Fokker-Planck equation (FPE) [21] with linear variational drift $g(y, z, t) = A(z, t)y + b(z, t)$ for every mode $z$ individually,

$$
\partial_t q_Y(y, t \mid z) = -\sum_{i=1}^n \partial_{y_i}\{g_i(y, z, t)q_Y(y, t \mid z)\} + \frac{1}{2}\sum_{i=1}^n\sum_{j=1}^n \partial_{y_i}\partial_{y_j}\{D_{ij}q_Y(y, t \mid z)\}. \qquad (14)
$$

Equation (14) describes the marginal density of a classical SDE, which, under linear drift, is equivalent to a GP [48]. This PDE is hence solved by a time-dependent Gaussian distribution $q_Y(y, t \mid z) = \mathcal{N}(y \mid \mu(z, t), \Sigma(z, t))$ [33], where the dynamics of the parameters is described by two ordinary differential equations (ODEs)

$$
\dot{\mu}(z, t) = A(z, t)\mu(z, t) + b(z, t), \quad \dot{\Sigma}(z, t) = A(z, t)\Sigma(z, t) + \Sigma(z, t)A^\top(z, t) + D, \ \forall z \in \mathcal{Z}. \quad (15)
$$

Our approach hence amounts to a mixture of GPs: this approximation will be accurate whenever the distribution $q_Z(z, t)$ is peaked at one $z \in \mathcal{Z}$. In this case, the HME separates into a FPE and a

master equation for the processes $Y(t) \mid \tilde{Z}$ and $Z(t)$, respectively; see Appendix A.3.2 for details. Accordingly, the approximation error over the whole interval $[0, T]$ will be small if the original system dynamics are linear in each mode and the modes are well discernible, that is, if the exact posterior concentrates on one mode. Since we are interested specifically in meta-stable systems which, by definition, transition between qualitatively different regimes and exhibit a separation of time scales between the intra-mode diffusive dynamics and the inter-mode transitions, we expect these criteria to be met reasonably well for our systems of interest.

The constraints (13) and (15) can be included into the objective Eq. (10) via Lagrange multiplier functions, yielding an augmented objective, the *Lagrangian*. We define the multipliers $\lambda(z, t)$, $\Psi(z, t), \nu(z, t)$ for the variational mean $\mu(z, t)$ and covariance $\Sigma(z, t)$ and variational rates $\tilde{\Lambda}(z, z', t)$, respectively. Writing the ELBO as $\mathcal{L}[\mathbb{Q}_{Y,Z}] = \int_0^T \ell_\mathbb{Q}(t) \, \mathrm{d}t$, the full Lagrangian $L$ to be maximized reads

$$
\begin{aligned}
L = \int_0^T \ell_\mathbb{Q}(t) + \sum_{z \in \mathcal{Z}} \Big[ \lambda^\top(z, t) \left( \dot{\mu}(z, t) - (A(z, t)\mu(z, t) + b(z, t)) \right) + \mathrm{tr} \Big\{ \Psi^\top(z, t) \Big( \dot{\Sigma}(z, t) \\
- \left( A(z, t)\Sigma(z, t) + \Sigma(z, t)A^\top(z, t) + D \right) \Big) \Big\} + \nu(z, t) \Big( \dot{q}_Z(z, t) - \sum_{z' \in \mathcal{Z}} \tilde{\Lambda}_{z'z}(t) q_Z(z', t) \Big) \Big] \, \mathrm{d}t,
\end{aligned}
\tag{16}
$$

where we used the shorthand $\tilde{\Lambda}_{z'z}(t) := \tilde{\Lambda}(z,' z, t)$ for $z' \neq z$ and $\tilde{\Lambda}_{zz}(t) := -\tilde{\Lambda}(z, t)$ else. Note that the dependency of $L$ on the variational measure $\mathbb{Q}_{Y,Z}$ is fully captured by the variational factors, $L = L[\mathbb{Q}_{Y,Z}] = L[q_Z, \mu, \Sigma]$.

## 3.2 Optimizing the Variational Distributions

The optimization problem consists in finding the optimal variational factors $q_Z^*, \mu^*, \Sigma^*$ and parameters $A^*, b^*, \tilde{\Lambda}^*, \phi^*$ maximizing Eq. (16), where $\phi$ summarizes the variational initial conditions. Our structured mean-field assumption Eq. (12) allows us to maximize Eq. (16) individually with respect to $q_Z(z, t)$ and $q_Y(y, t \mid z)$ [10], guaranteeing an increase in the ELBO due to convexity in each individual argument [25]. As a consequence of Pontryagin's maximum principle [49], the solutions to the maximization problems $q_Z^*(z, t) = \arg \max_{q_Z(z,t)} L$ and $q_Y^*(y, t \mid z) = \arg \max_{q_Y(y,t|z)} L$ have to fulfil the respective constraint equations (13) and (15) as well as the Euler-Lagrange (EL) equation $\frac{\mathrm{d}}{\mathrm{d}t} \partial_{\dot{q}} \ell = \partial_q \ell$, where $L = \int_0^T \ell(t) \, \mathrm{d}t$. The latter is giving rise to ODEs for the Lagrange multiplier functions: Firstly, the EL equation with respect to $q_Z(z, t)$ yields

$$
\dot{\nu}(z, t) = \partial_{q_Z(z,t)} \ell_\mathbb{Q} - \sum_{z' \in \mathcal{Z} \setminus z} \tilde{\Lambda}(z, z', t) \nu(z', t) + \tilde{\Lambda}(z, t) \nu(z, t). \tag{17}
$$

Secondly, the EL equations hold separately for both Gaussian parameters $\mu(z, t), \Sigma(z, t)$. We obtain

$$
\begin{aligned}
\dot{\lambda}(z, t) &= \partial_{\mu(z,t)} \ell_\mathbb{Q} - A^\top(z, t) \lambda(z, t), \\
\dot{\Psi}(z, t) &= \partial_{\Sigma(z,t)} \ell_\mathbb{Q} - A^\top(z, t) \Psi(z, t) - \Psi(z, t) A(z, t).
\end{aligned}
\tag{18}
$$

Both Eqs. (17) and (18) hold between observations; at the observation time points, reset conditions follow from the respective observation likelihoods. For the detailed derivations, including the explicit expressions for the gradients $\partial \ell_\mathbb{Q}$ for linear prior models $f(y, z, t) := A_p(z, t)y + b_p(z, t)$ as well as the reset conditions, see Appendix A.3.3. Note that this constitutes a set of *impulsive* ODEs scaling linearly in the number of observations [50]. For more details on the scaling behavior of these ODEs, see Appendix A.3.3. As the parameters of both Eq. (17) and Eq. (18) are time-dependent, general, analytic solutions does not exist [51]. Instead, we resort to established numerical solvers [52].

The full optimization problem requires the Lagrange multiplier ODEs (17) and (18) and the constraint Eqs. (13) and (15) to be solved jointly as a boundary-value problem with terminal conditions $\nu(\cdot, T), \lambda(\cdot, T), \Psi(\cdot, T) = 0$ and initial conditions on the distribution parameters [49]. The variational parameters have to be optimized simultaneously. A standard approach to this problem is an iterative forward-backward sweeping algorithm [53, 54]: (i) solve the Lagrange multiplier ODEs Eqs. (17) and (18) backward in time, starting from the terminal conditions $\nu, \lambda, \Psi = 0$. Next, (ii) update the

variational parameters acting on the constraints, in our case $A, b, \tilde{\Lambda}, \phi$. Here, we employ a simple gradient ascent scheme: for each $u(t) \in \{A(z, t), b(z, t), \tilde{\Lambda}(z, z', t), \phi\}$, we update

$$u(t) \leftarrow u(t) + \kappa(t) \cdot \partial_{u(t)} \ell, \tag{19}$$

where we use a back-tracking line search [55] for the step size $\kappa(t)$, see Appendix A.3.4. Then, (iii) solve the constraint equations (13) and (15) forward in time, starting from initial conditions $\phi = \{q_Z(z, 0) = q_Z^0(z), \mu(z, 0) = \mu^0(z), \Sigma(z, 0) = \Sigma^0(z)\}$. Finally, (iv) repeat until convergence.

Note that our results generalize the findings of [26, 29]: for $f(y, z, t) = f(y, t), g(y, z, t) = g(y, t)$, i.e., in the absence of coupling between the $Y$- and $Z$-processes, the MJP and diffusion contributions to the prior KL Eq. (11) separate and the derivative $\partial_{q_Z(z,t)} \ell_{\mathbb{Q}}$ reduces to the result of [29] between observations. Furthermore, for the special case $|\mathcal{Z}| = 1$, our result recovers the conventional GP approximation [26].

### 3.3 Parameter Learning

To learn the model parameters, that is, the prior transition rate matrix $\Lambda$, the dispersion $D$, the prior initial conditions, the parameters of the drift function $f(y, z, t)$ and the parameters of the observation likelihood $p(x_i \mid y_i)$, we employ a variational expectation maximization (VEM) scheme [10]. After converging onto variational distributions $q_Z(z, t)$ and $q_Y(y, t \mid z)$, we perform gradient ascent with respect to these parameters on the Lagrangian $L$, Eq. (16), see Appendix A.3.5. Note that we opt for this basic approach for simplicity so as to focus on the general inference framework. For an in-depth discussion on parameter optimization, see, e.g., [56]. The complete optimization scheme is summarized in Algorithm 1, where we subsume all model parameters under $\Theta$ for conciseness. This strategy yields a local optimum due to convexity in each individual argument [25]. Note that to alleviate potential issues with local optima it is straightforward to utilize, e.g., multi-start approaches [57].

---

**input :** observation data $\{t_i, x_i\}_{i=1,\dots,N}$

Initialize $q_Z, \mu, \Sigma, A, b, \tilde{\Lambda}, \Theta$
**while** $\mathcal{L}$ *not converged* **do**
    **while** $\mathcal{L}$ *not converged* **do**
        Compute multiplier functions $\lambda$, $\Psi, \nu$ via Eqs. (17) and (18)
        Update variational parameters $A$, $b, \tilde{\Lambda}$ via Eq. (19)
        Compute variational factors $\mu, \Sigma$, $q_Z$ via Eqs. (13) and (15)
        Update lower bound $\mathcal{L}$
    **end**
    Update prior parameters $\Theta$ via gradient ascent
    Update lower bound $\mathcal{L}$
**end**

---

**Algorithm 1:** VI for hybrid processes

## 4 Experiments

### 4.1 Model Validation on Ground-Truth Data

We validate our method on synthetic data generated from a 1D, two-mode hybrid system with observations corrupted by Gaussian noise, $p(x_i \mid y_i) = \mathcal{N}(x_i \mid y_i, \Sigma_{\text{obs}})$, where the observation times are drawn from a Poisson point process [20]. Both mode dynamics are given by time-independent linear drift functions

$$f(y, z, t) = \alpha_z(\beta_z - y), \tag{20}$$

with set points $\beta_z$ and dynamics $\alpha_z > 0$, $z \in \mathcal{Z} = \{1, 2\}$. For $|\mathcal{Z}| = 1$, this would recover the well-known Ornstein-Uhlenbeck process [33].

As shown in Fig. 2 A, the inferred posterior distributions $q_Y(y, t) = \sum_{z \in \mathcal{Z}} q_Y(y, t \mid z)$ and $q_Z(z, t)$ both faithfully reconstruct the respective latent ground-truth trajectories. This is also reflected by the maximum a-posteriori (MAP) paths $y^{\text{MAP}}(t), z^{\text{MAP}}(t) = \arg \max_{y,z} q(y, z, t)$. In regions around mode transitions, one can observe artifacts from the variational approximation as a mixture of GPs: in the time interval $\Delta t_{\text{trans}}$ between the last observation before and the first observation after the ground-truth mode transition at $t \approx 12.5$, the marginal $q_Y(y, t)$ does not exhibit a smooth transition across $Y = 0$, but splits the probability density between the independently evolving Gaussian distributions $q_Y(y, t \mid z)$. This is not surprising, as we have argued (c.f. Section 3.1) that our approximation will be accurate in regions where $q_Z(z, t) \approx 1$, which is the case at the beginning and the end of the interval $\Delta t_1$, but not in between. Since the relaxation onto the mode set points $\beta_z$ is fast compared to the mode

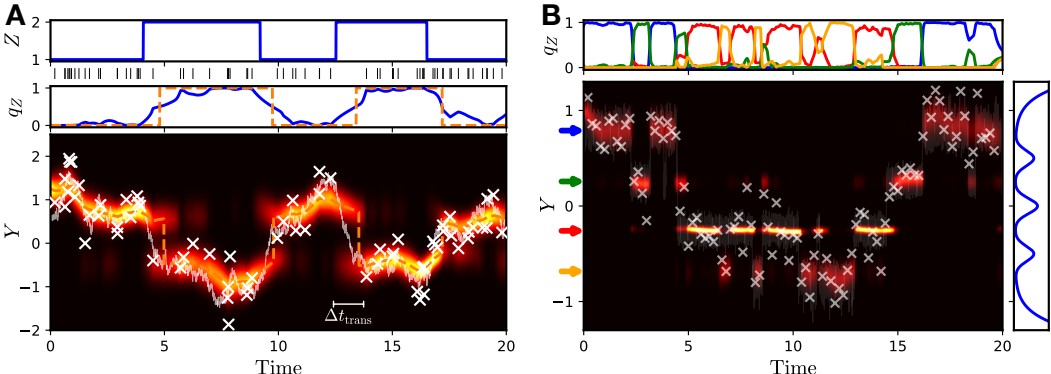

Figure 2: **A**: Model validation on ground-truth data from a 1D, two-mode hybrid process. Top: True discrete path $z(t)$ with observation times (vertical lines below). Middle: inferred marginal $q_Z(2, t)$ and $z^{\mathrm{MAP}}(t)$ (dashed). Bottom: true SSDE path (gray line), observations (crosses), the inferred marginal $q_Y(y, t)$ and $y^{\mathrm{MAP}}(t)$ (dashed). Brighter colors indicate higher probability density. **B**: Diffusion in a 1D four-well potential. Top: inferred marginals $q_Z(z, t)$. Bottom: true SDE and observations with the inferred $q_Y(y, t)$ and learned set points $\beta_z$ (arrows). Right: potential landscape.

remain times, these transition regions are short, yielding a high approximation quality. This is further highlighted by comparison with an experiment in which this condition is violated, which we show in Appendix B.1. As we pursue a generative modeling approach, we can verify the approximation quality by sampling full trajectories from the variational posterior. The empirical distribution over paths closely resembles the latent continuous trajectory. We provide a plot of the sampling distribution in Appendix B.1 along with a version of Fig. 2 A showing the dynamics of the individual modes. Furthermore, the model parameters are identified with high accuracy: the learned set points, for instance, $\beta_1 = 0.70, \beta_2 = -0.52$, where the ground-truth values are $\pm 1$. The exhaustive list of both the learned and ground truth model parameters is also provided in Appendix B.1.

## 4.2 Diffusions in Multi-Well Potentials

In many real-world scenarios, ground-truth discrete modes driving continuous dynamics do not exist, but continuous dynamics often exhibit qualitatively different regimes. Transitions between different regimes typically occur on vastly longer time scales than the relaxation dynamics within each regime, as is observed, e.g., for the folding dynamics of complex biomolecules [58]. For such meta-stable systems in particular, explicit probabilistic modeling of a set of underlying discrete modes can greatly aid interpretability and enable targeted interventions on the system. To demonstrate the capability of our model to yield sensible representations of distinct dynamic regimes, we apply it to latent Itô diffusions driven by 1D and 2D benchmark potentials widely used in computational biology [59, 60, 61, 62] We model these system via hybrid processes with linear drift, c.f. Eq. (20), and Gaussian observation noise in both the 1D and 2D case, as before. We assume a mode-dependent dispersion, $D = D(z)$. The observation time points are regularly spaced and we fix the observation covariance $\Sigma_{\mathrm{obs}}$.

In both the 1D and 2D case, the mode reconstructions accurately capture the global transitions between distinct potential minima, see Fig. 2 B and Fig. 3 B. We note that in the 1D example (Fig. 2 B), the true latent continuous trajectory exhibits a particularly clear separation of time scales between the inter- and intra-well dynamics, which is reflected in sharp transitions in the mode reconstruction and accordingly particularly accurate results. On the other hand, as shown in Fig. 3, in the 2D case, more pronounced transition regions exist, where it is not possible to unambiguously assign the state at a given time $t$ to one of the three minima. The posterior marginals $q_Z(z, t)$ sensibly capture this uncertainty, which is also reflected in a high quality mode-assignment of the observed data points $x_{[1,N]}$ as shown in Fig. 3 A. Furthermore, the asymmetry of the learned mode-dependent dispersions accurately reflects the topology of the underlying potential. An overview over all parameters is given in Appendix B.2.

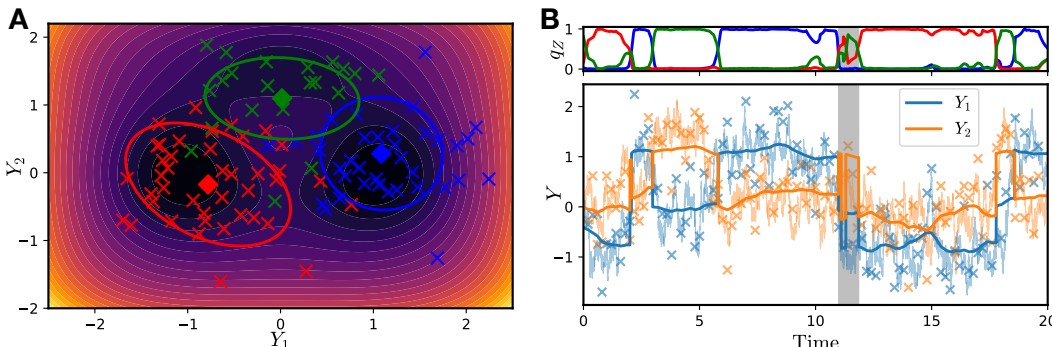

Figure 3: Diffusion in a three-well potential. **A**: Potential landscape with the inferred set points $\beta_z$ (diamonds) and dispersions $D(z)$ (ellipses, $3\sigma$-region) with observations (crosses); colors according to $z^{\mathrm{MAP}}(t_i)$ for each observation $x_i$. **B**: Top: inferred marginals $q_Z(z, t)$. Bottom: components of $y^{\mathrm{MAP}}(t)$ (thick lines), the ground-truth path (thin lines) and the observations (crosses). Shaded region: transition region with high ambiguity.

## 4.3 Switching Ion Channel Data

We apply our method to a structural molecular biology problem: we aim to identify the switching behavior of the viral ion channel Kcv$_{\mathrm{MT325}}$ exhibiting three different channel conformations [63]. Different conformations yield different ion permeabilities and hence different conductivities which can be directly detected by applying a voltage across the cell membrane and measuring the trans-membrane current. We model the conformation dynamics as discrete process $Z$ and the current, passing through an analog (continuous-time) filter and incurring amplifier noise, as continuous process $Y$. The observations $X_i$ are sampled at a fixed rate and subject to quantization errors from an analog-to-digital converter. The observation noise is a known property of the used setup; we hence fix $\Sigma_{\mathrm{obs}}$. We reconstruct a highly plausible switching behavior and filter out individual outliers, as depicted in Fig. 4; see Appendix B.3 for a list of all learned parameters and experimental details. We note that very similar problem setups can be found, e.g., in nanopore sequencing technologies [64].

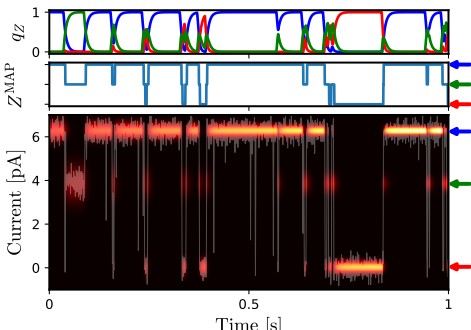

Figure 4: Switching behavior of viral ion channel Kcv$_{\mathrm{MT325}}$. Top: marginals $q_Z(z, t)$ corresponding to three channel conformations. Middle: $z^{\mathrm{MAP}}(t)$. Bottom: data (white line), marginals $q_Y(y, t)$ and learned set points $\beta_z$ (arrows). Brighter colors indicate higher probability density.

## 4.4 Learning Complex Latent Continuous Dynamics

After having demonstrated the applicability of our model to hybrid systems with different time scales for the discrete and continuous dynamics, we lastly show that it also works well when this criterion is not met. In areas such as automation and robotics, hybrid models are ubiquitously used to, e.g., encode highly complex continuous movements via a discrete set of movement primitives [65], which are non-stationary processes. To demonstrate that our method is able to reconstruct such complex latent continuous dynamics, we employ a 2D version of Eq. (20) where the mode dynamics $f(y, z, t) = f(y, z)$ are given as two counter-rotating vector fields, see Fig. 5 B. We fix the observation covariance, as we can assume its value to be known and small compared to the system volatility for applications such as robotics.

As shown in Fig. 5 A and C, the mode and state reconstructions accurately recover the true paths. Accordingly, also the underlying mode dynamics are correctly learned, exhibiting the counter-rotating behavior of the ground-truth model. We provide a list of all parameters in Appendix B.4.

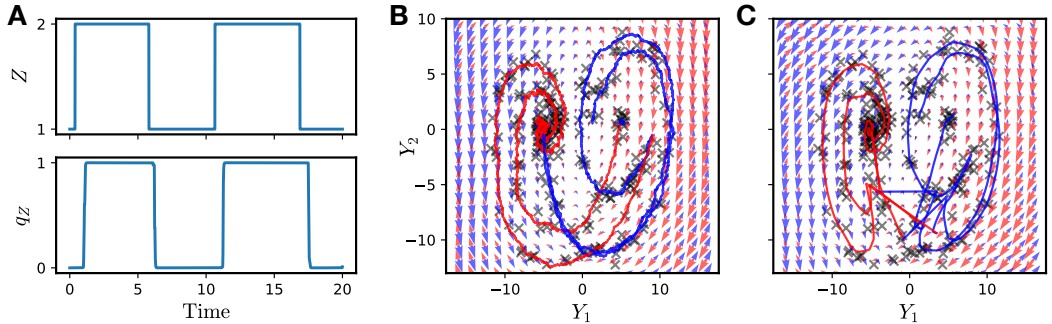

Figure 5: Inference of complex structured continuous dynamics. **A**: True mode path $z(t)$ (top) and inferred marginal $q_Z(2, t)$ (bottom). **B**: True state path $y(t)$ (coloring according to associated mode; $Z = 1$: blue, $Z = 2$: red), observations (crosses) and true mode dynamics $f(y, z)$ (arrows) in the phase plane. **C**: Reconstructed MAP path $y^{\mathrm{MAP}}(t)$ (coloring according to $z^{\mathrm{MAP}}(t)$) and reconstructed mode dynamics (arrows).

## 5   Conclusion

We presented, to the best of our knowledge, the first variational inference framework for continuous time hybrid process models: since the exact filtering and smoothing distributions are computationally intractable, we proposed a variational approximation to the exact model. The key assumption is that the true discrete posterior is peaked at any $z \in \mathcal{Z}$ for extended periods of time, allowing for a straightforward, easily interpretable mixture of GPs to be used as approximation. We have evaluated our framework on various benchmark tasks including real-world biological data and demonstrated its ability to faithfully reconstruct complex latent dynamics and to learn the unknown system parameters, in particular in applications to meta-stable systems. While we implemented parameter learning via point estimates, we aim to extend this to a fully Bayesian framework in the future, enabling the integration of prior domain knowledge and the associated uncertainty about the system at hand. Furthermore, due to the scaling behavior of the ODEs (13) and (15), additional approximations need to be worked out to be able to apply the method to high-dimensional state spaces and large numbers of modes, see, e.g., [66]. As many natural and engineered systems can be described as hybrid systems, we think that extending the toolbox for inference will be of great utility for the analysis and control of such systems.

### Acknowledgments

We thank Gerhard Thiel and Kerri Kukovetz for providing the ion channel voltage data and helpful discussions and the anonymous reviewers for their useful comments and suggestions. This work has been funded by the German Research Foundation (DFG) as part of the project B4 within the Collaborative Research Center (CRC) 1053 – MAKI, and by the European Research Council (ERC) within the CONSYN project, grant agreement number 773196.

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
