# Variational Inference for Continuous-Time Switching Dynamical Systems
## — Supplementary Material —

**Lukas Köhs**     **Bastian Alt**     **Heinz Koeppl**
Department of Electrical Engineering and Information Technology
Technische Universität Darmstadt
{lukas.koehs, bastian.alt, heinz.koeppl}@bcs.tu-darmstadt.de

## A   Derivations

For the derivations we will use the notation

$$p(x) := \partial_x \mathbb{P}(X \leq x)$$

for the density function of a random variable $X$. For the conditional density of a random variable $X$ and a realization $y$ of a random variable $Y$, we write

$$p(x \mid y) := \partial_x \mathbb{P}(X \leq x \mid Y = y).$$

For time-dependent densities with continuous random variables $Y(t)$ and discrete random variables $Z(t)$, we use the time-point marginal density

$$p(y, z, t) := \partial_y \mathbb{P}(Y(t) \leq y, Z(t) = z),$$

time-point joint density

$$p(y, z, t, y', z', t') = \partial_y \partial_{y'} \mathbb{P}(Y(t) \leq y, Z(t) = z, Y(t') \leq y', Z(t') = z')$$

and time-point conditional density

$$p(y, z, t \mid y', z', t') = \partial_y \mathbb{P}(Y(t) \leq y, Z(t) = z \mid Y(t') = y', Z(t') = z').$$

The latter also applies to conditional densities with multiple time points in the conditioning set,

$$
\begin{aligned}
&p(y, z, t \mid y', z', t', y'', z'', t'') \\
&:= \partial_y \mathbb{P}(Y(t) \leq y, Z(t) = z \mid Y(t') = y', Z(t') = z', Y(t'') = y'', Z(t'') = z'').
\end{aligned}
$$

If it is clear from the context, we will mostly use the favorable uncluttered notation.

### A.1   Derivation of the Hybrid Master Equation

To derive the HME conditioned on an arbitrary set $\mathcal{X}$, e.g., the set of initial conditions $\mathcal{X} = \{Z(0) = z_0, Y(0) = y_0\}$, we assume for simplicity that $Z(t) \in \mathcal{Z} \subseteq \mathbb{N}$ and $Y(t) \in \mathcal{Y} \subseteq \mathbb{R}$. The multivariate case $\mathcal{Y} \subseteq \mathbb{R}^n$ is be derived analogously.

Following [34], we use the rule of total probability on the density $p(y, z, t + h \mid \mathcal{X})$ for some $h > 0$:

$$
\begin{aligned}
p(y, z, t + h \mid \mathcal{X}) &= \sum_{z'} \int_{\mathcal{Y}} p(y, z, t + h \mid y', z', t, \mathcal{X}) p(y', z', t \mid \mathcal{X}) \, \mathrm{d}y' \\
&= \sum_{z'} \int_{\mathcal{Y}} p(y, t + h \mid z, t + h, y', z', t, \mathcal{X}) p(z, t + h \mid y', z', t, \mathcal{X}) \\
&\qquad\qquad \cdot p(y', z', t \mid \mathcal{X}) \, \mathrm{d}y'.
\end{aligned}
$$

We expand, with $\lim_{h\to 0}\frac{1}{h}p(z,t+h\mid y',z',t,\mathcal{X}):=\Lambda^{y'}_{z'z}(t)$ and $\lim_{h\to 0}\frac{o(h)}{h}=0$,

$$p(z,t+h\mid y',z',t,\mathcal{X})=\delta_{z'z}+\Lambda^{y'}_{z'z}(t)h+o(h)$$

with the Kronecker delta $\delta_{z'z}=1$ if $z'=z$ and 0 otherwise. Since we aim to take the limit $h\to 0$ at the end, we omit terms of $o(h)$. Inserting the expansion into the above expression, we obtain

$$
\begin{aligned}
&p(y,z,t+h\mid\mathcal{X})\\
&=\sum_{z'}\int_{\mathcal{Y}}p(y,t+h\mid z,t+h,y',z',t,\mathcal{X})\left(\delta_{z'z}+\Lambda^{y'}_{z'z}(t)h\right)p(y',z',t\mid\mathcal{X})\,\mathrm{d}y'\\
&=\int_{\mathcal{Y}}p(y,t+h\mid z,t+h,y',z,t,\mathcal{X})p(y',z,t\mid\mathcal{X})\,\mathrm{d}y'\\
&\quad+\sum_{z'}\int_{\mathcal{Y}}p(y,t+h\mid z,t+h,y',z',t,\mathcal{X})\Lambda^{y'}_{z'z}(t)hp(y',z',t\mid\mathcal{X})\,\mathrm{d}y'
\end{aligned}
\tag{21}
$$

The density function $p(y,t+h\mid z,t+h,y',z',t,\mathcal{X})$ can be written in terms of its characteristic function $\psi(\nu,t+h\mid y',t,\mathcal{X})=\mathsf{E}\left[e^{i\nu(Y(t+h)-Y(t))}\mid Z(t+h)=z,Y(t)=y',Z(t)=z',\mathcal{X}\right]$,

$$
\begin{aligned}
&p(y,t+h\mid z,t+h,y',z',t,\mathcal{X})\\
&=\frac{1}{2\pi}\int_{\mathbb{R}}e^{-i\nu(y-y')}\psi(\nu,t+h\mid y',t,\mathcal{X})\,\mathrm{d}\nu\\
&=\frac{1}{2\pi}\int_{\mathbb{R}}e^{-i\nu(y-y')}\\
&\quad\cdot\sum_{n=0}^{\infty}\frac{(i\nu)^n}{n!}\,\mathsf{E}\left[(Y(t+h)-Y(t))^n\mid Z(t+h)=z,Y(t)=y',Z(t)=z',\mathcal{X}\right]\mathrm{d}\nu,
\end{aligned}
\tag{22}
$$

where we expressed the characteristic function via its Taylor series around $\nu=0$. We insert this representation into Eq. (21) and make use of the identity (which only holds under the integral)

$$\partial_y^{(n)}\delta(y-y')=\frac{1}{2\pi}(-i\nu)^n\int_{\mathbb{R}}e^{-i\nu(y-y')}\,\mathrm{d}\nu,$$

with $\partial_y^{(0)}\delta(y-y'):=\delta(y-y')$, yielding

$$
\begin{aligned}
&p(y,z,t+h\mid\mathcal{X})\\
&=\int_{\mathcal{Y}}p(y,t+h\mid z,t+h,y',z,t,\mathcal{X})p(y',z,t\mid X)\,\mathrm{d}y'\\
&\quad+h\cdot\sum_{z'}\int_{\mathcal{Y}}p(y,t+h\mid z,t+h,y',z',t,\mathcal{X})\Lambda^{y'}_{z'z}(t)p(y',z',t\mid\mathcal{X})\,\mathrm{d}y'\\
&=\int_{\mathcal{Y}}\sum_{n=0}^{\infty}\frac{(-1)^n}{n!}\partial_y^{(n)}\delta(y-y')\,\mathsf{E}\left[(Y(t+h)-Y(t))^n\mid Z(t+h)=z,\right.\\
&\hspace{5cm}\left.Y(t)=y',Z(t)=z,\mathcal{X}\right]p(y',z,t\mid\mathcal{X})\,\mathrm{d}y'\\
&\quad+h\cdot\sum_{z'}\int_{\mathcal{Y}}\sum_{n=0}^{\infty}\frac{(-1)^n}{n!}\partial_y^{(n)}\delta(y-y')\,\mathsf{E}\left[(Y(t+h)-Y(t))^n\mid Z(t+h)=z,\right.\\
&\hspace{4cm}\left.Y(t)=y',Z(t)=z',\mathcal{X}\right]\Lambda^{y'}_{z'z}(t)p(y',z',t\mid\mathcal{X})\,\mathrm{d}y'\\
&=\sum_{n=0}^{\infty}\frac{(-1)^n}{n!}\partial_y^{(n)}\,\mathsf{E}\left[(Y(t+h)-Y(t))^n\mid Z(t+h)=z,Y(t)=y,Z(t)=z,\mathcal{X}\right]p(y,z,t\mid\mathcal{X})\\
&\quad+h\cdot\sum_{z'}\sum_{n=0}^{\infty}\frac{(-1)^n}{n!}\partial_y^{(n)}\,\mathsf{E}\left[(Y(t+h)-Y(t))^n\mid Z(t+h)=z,Y(t)=y,Z(t)=z',\mathcal{X}\right]\\
&\hspace{6cm}\cdot\Lambda^{y}_{z'z}(t)p(y,z',t\mid\mathcal{X}).
\end{aligned}
$$

We again apply a Taylor expansion and omit part of the conditioning set for brevity,

$$\mathsf{E}\left[(Y(t+h)-Y(t))^n|Y(t)=y\right] = \sum_{m=0}^{\infty}\frac{h^m}{m!}\partial_{\tau}^{(m)}\mathsf{E}\left[(Y(t+\tau)-Y(t))^n|Y(t)=y\right]|_{\tau=0}$$
$$= 1 + o(1).\qquad(23)$$

Inserting this and omitting terms $o(h)$, we obtain

$$p(y,z,t+h\mid\mathcal{X})$$
$$= \sum_{n=0}^{\infty}\frac{(-1)^n}{n!}\partial_y^{(n)}\mathsf{E}\left[(Y(t+h)-Y(t))^n|Z(t+h)=z,Y(t)=y,Z(t)=z,\mathcal{X}\right]p(y,z,t\mid\mathcal{X})$$
$$+ h\cdot\sum_{z'}\Lambda_{z'z}^y(t)p(y,z',t\mid\mathcal{X})$$
$$= p(y,z,t\mid\mathcal{X})+$$
$$\sum_{n=1}^{\infty}\frac{(-1)^n}{n!}\partial_y^{(n)}\mathsf{E}\left[(Y(t+h)-Y(t))^n|Z(t+h)=z,Y(t)=y,Z(t)=z,\mathcal{X}\right]p(y,z,t\mid\mathcal{X})$$
$$+ h\cdot\sum_{z'}\Lambda_{z'z}^y(t)p(y,z',t\mid\mathcal{X})$$

Substracting $p(y,z,t\mid\mathcal{X})$ from both sides, dividing by $h$ and taking the limit $h\to0$ yields

$$\partial_t p(y,z,t\mid\mathcal{X}) = \lim_{h\to0}\frac{p(y,z,t+h\mid\mathcal{X})-p(y,z,t\mid\mathcal{X})}{h}$$
$$= \sum_{n=1}^{\infty}\frac{(-1)^n}{n!}\partial_y^{(n)}\{\Gamma_{nyz}p(y,z,t\mid\mathcal{X})\} + \sum_{z'}\Lambda_{z'z}^y(t)p(y,z',t\mid\mathcal{X})\qquad(24)$$

with

$$\Gamma_{nyz} = \lim_{h\to0}\frac{1}{h}\mathsf{E}\left[(Y(t+h)-Y(t))^n|Z(t+h)=z,Y(t)=y,Z(t)=z,\mathcal{X}\right]$$
$$\Lambda_{z'z}^y = \lim_{h\to0}\frac{1}{h}p(z,t+h\mid z',y',t,\mathcal{X})-\delta_{z'z}.\qquad(25)$$

As $Y(t)$ follows the SSDE

$$\mathrm{d}Y(t) = f(Y(t),Z(t),t)\,\mathrm{d}t + Q(Y(t),Z(t),t)\,\mathrm{d}W(t),$$

we can compute the conditional moments $\Gamma_{nyz}$ in closed form. Conditioned on the discrete process remaining constant in a small time interval, $Z_{[t,t+h]}=z$, the above SSDE can be treated as a conventional, $Z$-independent Itô SDE. For small $h$, we can hence utilize the usual Euler-Maruyama approximation [33],

$$Y(t+h)|Z(t+h)=z,Z(t)=z,Y(t)=y \sim \mathcal{N}\left(y+f(y,z,t)h,D(y,z,t)h\right).$$

Consequently,

$$\mathsf{E}\left[(Y(t+h)-Y(t))^n|Z(t+h)=z,Y(t)=y,Z(t)=z,\mathcal{X}\right]$$
$$= \int(y'-y)^n\mathcal{N}\left(y+f(y,z,t)h,D(y,z,t)h\right)\mathrm{d}y'$$

and the first two conditional moments are the usual Gaussian moments

$$\Gamma_{nyz} = \begin{cases} f(y,z,t) & \text{if } n=1 \\ \frac{1}{2}Q(y,z,t)Q^\top(y,z,t) = \frac{1}{2}D(y,z,t) & \text{if } n=2. \end{cases}$$

As shown in [34], if $\Gamma_{nyz}=0$ for some even $n$, $\Gamma_{nyz}=0\,\forall n\geq0$. It is straightforward to show, e.g., that $\Gamma_{nyz}=0$ for $n=4$, so all other conditional moments vanish. Hence, we can (for arbitrary $\mathcal{Y}\subseteq\mathbb{R}^n$) define the PDE

$$\partial_t p(y,z,t\mid\mathcal{X}) = \mathcal{A}p(y,z,t\mid\mathcal{X})$$

using the operator $\mathcal{A}(\cdot) = \mathcal{F}(\cdot) + \mathcal{T}(\cdot)$ as

$$\mathcal{F}p(y,z,t \mid \mathcal{X}) = -\sum_{i=1}^{n} \partial_{y_i} \{f_i(y,z,t)p(y,z,t \mid \mathcal{X})\}$$

$$+ \frac{1}{2}\sum_{i=1}^{n}\sum_{j=1}^{n} \partial_{y_i}\partial_{y_j}\{D_{ij}(y,z,t)p(y,z,t \mid \mathcal{X})\},$$

$$\mathcal{T}p(y,z,t \mid \mathcal{X}) = \sum_{z' \in \mathcal{Z}\backslash z} \Lambda(z',z,t)p(y,z',t \mid \mathcal{X}) - \Lambda(z,t)p(y,z,t \mid \mathcal{X}).$$

In the same vein as the above derivation, using the Kolmogorov backward equation

$$p(\mathcal{X} \mid y,z,t-h) = \sum_{z' \in \mathcal{Z}} \int p(y',z',t \mid y,z,t-h)p(\mathcal{X} \mid y',z',t)\,\mathrm{d}y',$$

we can find another PDE for the density $p(\mathcal{X} \mid y,z,t)$. This yields the backward equation $\partial_t p(\mathcal{X} \mid y,z,t) = -\mathcal{A}^\dagger p(\mathcal{X} \mid y,z,t)$, with the adjoint operator $\mathcal{A}^\dagger(\cdot) = \mathcal{F}^\dagger(\cdot) + \mathcal{T}^\dagger(\cdot)$:

$$\mathcal{F}^\dagger p(\mathcal{X} \mid y,z,t) = \sum_{i=1}^{n} f_i(y,z,t)\partial_{y_i}p(\mathcal{X} \mid y,z,t) + \frac{1}{2}\sum_{i=1}^{n}\sum_{j=1}^{n} D_{ij}(y,z,t)\partial_{y_i}\partial_{y_j}p(\mathcal{X} \mid y,z,t),$$

$$\mathcal{T}^\dagger p(\mathcal{X} \mid y,z,t) = \sum_{z' \in \mathcal{Z}\backslash z} \Lambda(z,z',t)p(\mathcal{X} \mid y,z,t) - \Lambda(z,t)p(\mathcal{X} \mid y,z,t).$$

The operator $\mathcal{A}^\dagger$ is adjoint to the operator $\mathcal{A}$, with respect to the inner product $\langle p,\phi \rangle := \sum_z \int p(y,z,t)\phi(y,z,t)\,\mathrm{d}y$, i.e.

$$\langle \mathcal{A}p, \phi \rangle = \langle p, \mathcal{A}^\dagger\phi \rangle$$

for an arbitrary test function $\phi$.

### A.1.1 Exact Marginal $Z$-Process

We here show that integrating out the continuous variable $y$ from the HME yields the traditional master equation. The full HME reads

$$\partial_t p(y,z,t) = \mathcal{A}p(y,z,t)$$

$$= -\sum_{i=1}^{n} \partial_{y_i}\{f_i(y,z,t)p(y,z,t)\} + \frac{1}{2}\sum_{i=1}^{n}\sum_{j=1}^{n} \partial_{y_i}\partial_{y_j}\{D_{ij}(y,z,t)p(y,z,t)\}$$

$$+ \sum_{z' \in \mathcal{Z}\backslash z} \Lambda(z',z,t)p(y,z',t) - \Lambda(z,t)p(y,z,t).$$

Using Leibniz' theorem, we have

$$\int_{\mathcal{Y}} \partial_t p(y,z,t)\mathrm{d}y = \partial_t \int_{\mathcal{Y}} p(y,z,t)\mathrm{d}y$$

$$= \partial_t p(z,t).$$

Accordingly, we have

$$\partial_t p(z,t) = \underbrace{\int_{\mathcal{Y}}\left(-\sum_{i=1}^{n}\partial_{y_i}\{f_i(y,z,t)p(y,z,t)\} + \frac{1}{2}\sum_{i=1}^{n}\sum_{j=1}^{n}\partial_{y_i}\partial_{y_j}\{D_{ij}(y,z,t)p(y,z,t)\}\right)\mathrm{d}y}_{=0}$$

$$+ \underbrace{\int_{\mathcal{Y}}\sum_{z' \in \mathcal{Z}\backslash z}\Lambda(z',z,t)p(y,z',t) - \Lambda(z,t)p(y,z,t)\,\mathrm{d}y}_{=\sum_{z' \in \mathcal{Z}\backslash z}\Lambda(z',z,t)p(z',t)-\Lambda(z,t)p(z,t)}$$

$$(26)$$

and the first integral has to vanish because of the Gauss divergence theorem and $p(y, z, t) \xrightarrow{y \to 0} 0$. Hence,

$$\partial_t p(z, t) = \sum_{z' \in \mathcal{Z} \setminus z} \Lambda(z', z, t) p(z', t) - \Lambda(z, t) p(z, t).$$

## A.2 Exact Posterior Inference

Here, we show how to calculate the quantities related to the smoothing density $p(y, z, t \mid x_{[1,N]}) := \partial_{y_1} \cdots \partial_{y_n} \mathbb{P}(Y(t) \leq y, Z(t) = z \mid X_1 = x_1, \ldots, X_N = x_N)$. Using $k = \max(k' \in \mathbb{N} \mid t_{k'} \leq t)$ we can be express the smoothing density as

$$
\begin{aligned}
p(y, z, t \mid x_{[1,N]}) &= \frac{p(y, z, t, x_1, \ldots, x_k, x_{k+1}, \ldots, x_N)}{p(x_1, \ldots, x_k, x_{k+1}, \ldots, x_N)} \\
&= \frac{p(x_{k+1}, \ldots, x_N \mid x_1, \ldots, x_k, y, z, t)}{p(x_{k+1}, \ldots, x_N \mid x_1, \ldots, x_k)} \frac{p(y, z, t, x_1, \ldots, x_k)}{p(x_1, \ldots, x_k)} \\
&= \frac{p(x_{k+1}, \ldots, x_N \mid y, z, t)}{p(x_{k+1}, \ldots, x_N \mid x_1, \ldots, x_k)} p(y, z, t \mid x_1, \ldots, x_k) \\
&= C^{-1}(t) \alpha(y, z, t) \beta(y, z, t),
\end{aligned}
$$

with the filtering density $\alpha(y, z, t) = p(y, z, t \mid x_1, \ldots, x_k)$, the backward density $\beta(y, z, t) = p(x_{k+1}, \ldots, x_N \mid y, z, t)$ and a time-dependent normalizer $C(t) = \sum_z \int \alpha(y, z, t) \beta(y, z, t) \, \mathrm{d}y$.

### A.2.1 Calculation of the Filtering Distribution

The filtering distribution is defined as

$$\alpha(y, z, t) := p(y, z, t \mid x_1, \ldots, x_k),$$

with density $p(y, z, t \mid x_1, \ldots, x_k) := \partial_{y_1} \cdots \partial_{y_n} \mathbb{P}(Y(t) \leq y, Z(t) = z \mid X_1 = x_1, \ldots, X_k = x_k)$ and $k = \max(k' \in \mathbb{N} \mid t_{k'} \leq t)$.

**The Filtering Distribution Between Observations.** Consider the case where there is no observation in the interval $[t, t + h]$, $h > 0$.

We compute

$$
\begin{aligned}
\alpha(y, z, t + h) &= p(y, z, t + h \mid x_1, \ldots, x_k) \\
&= \sum_{z' \in \mathcal{Z}} \int p(y, z, t + h, y', z', t \mid x_1, \ldots, x_k) \, \mathrm{d}y' \\
&= \sum_{z' \in \mathcal{Z}} \int p(y, z, t + h \mid y', z', t, x_1, \ldots, x_k) p(y', z', t \mid x_1, \ldots, x_k) \, \mathrm{d}y'.
\end{aligned}
$$

As there are no observations in the interval $[t, t + h]$, we have

$$p(y, z, t + h \mid y', z', t, x_1, \ldots, x_k) = p(y, z, t + h \mid y', z', t).$$

This is true since the conditional process $\{Y(t + h), Z(t + h)\}$ given $\{Y(t), Z(t)\}$ is independent of $\{X_1, \ldots, X_k\}$. Hence, we have

$$\alpha(y, z, t + h) = \sum_{z' \in \mathcal{Z}} \int p(y, z, t + h \mid y', z', t) \alpha(y', z', t)) \, \mathrm{d}y'.$$

This is the (forward) Chapman-Kolmogorov equation [34] for the filtering process $\{Y(t), Z(t) \mid x_1, \ldots, x_k\}$, with transition distribution $p(y, z, t + h \mid y', z', t)$, which is the transition distribution of the prior dynamics. Hence, between observations $\alpha(y, z, t)$ follows the HME

$$\partial_t \alpha(y, z, t) = \mathcal{A} \alpha(y, z, t),$$

as derived in Appendix A.1.

**The Filtering Distribution at Observation Time Points.**    Here, we calculate the filtering distribution at the observation time points $\{t_i\}_{i \in 1,\ldots,N}$.

$$\alpha(y, z, t_i) = p(y, z, t_i \mid x_1, \ldots, x_i)$$
$$= \frac{p(y, z, t_i, x_1, \ldots, x_i)}{p(x_1, \ldots, x_i)}$$
$$= \frac{p(x_i \mid y, z, t_i, x_1, \ldots, x_{i-1})p(y, z, t_i, x_1, \ldots, x_{i-1})}{p(x_1, \ldots, x_i)}$$
$$= \frac{p(x_i \mid y, z, t_i, x_1, \ldots, x_{i-1})p(y, z, t_i \mid x_1, \ldots, x_{i-1})p(x_1, \ldots, x_{i-1})}{p(x_1, \ldots, x_i)}$$
$$= \frac{p(x_i \mid y)\alpha(y, z, t_i^-)}{\tilde{C}_i}$$

and $\tilde{C}_i = \frac{p(x_1, \ldots, x_i)}{p(x_1, \ldots, x_{i-1})} = \sum_{z \in \mathcal{Z}} \int p(x_i \mid y)\alpha(y, z, t_i^-)\,\mathrm{d}y$.

### A.2.2    Calculation of the Backward Distribution

The backward distribution is defined as

$$\beta(y, z, t) := p(x_{k+1}, \ldots, x_N \mid y, z, t),$$

with density $p(x_{k+1}, \ldots, x_N \mid y, z, t) := \partial_{x_{k+1}} \cdots \partial_{x_N} \mathbb{P}(X_{n+1} \leq x_{n+1}, \ldots, X_N \leq x_N \mid Z(t) = y, Z(t) = z)$ and $k = \max(k' \in \mathbb{N} \mid t_{k'} \leq t)$

**The Backward Distribution Between Observations.**    Consider again first the case where there is no observation in the interval $[t - h, t]$, $h > 0$.

$$\beta(y, z, t - h) = p(x_{k+1}, \ldots, x_N \mid y, z, t - h)$$
$$= \sum_{z' \in \mathcal{Z}} \int p(x_{k+1}, \ldots, x_N, y', z', t \mid y, z, t - h)\,\mathrm{d}y'$$
$$= \sum_{z' \in \mathcal{Z}} \int p(y', z', t \mid y, z, t - h)p(x_{k+1}, \ldots, x_N \mid y', z', t, y, z, t - h)\,\mathrm{d}y'.$$

As there are no observations in the interval $[t - h, t]$,

$$p(x_{k+1}, \ldots, x_N \mid y', z', t, y, z, t - h) = p(x_{k+1}, \ldots, x_N \mid y', z', t) = \beta(y', z', t)$$

as the process $\{x_{k+1}, \ldots, x_N \mid Y(t), Z(t)\}$ is independent of $\{Y(t - h), Z(t - h)\}$. Hence,

$$\beta(y, z, t - h) = \sum_{z' \in \mathcal{Z}} \int p(y', z', t \mid y, z, t - h)\beta(y', z', t)\,\mathrm{d}y'.$$

This is the (backward) Chapman-Kolmogorov equation [34] for the backward process $\{x_{k+1}, \ldots, x_N \mid Y(t), Z(t)\}$, with transition distribution $p(y', z', t \mid y, z, t - h)$, which corresponds to the backward prior dynamics. Hence, between observation $\beta(y, z, t)$ follows the backward HME

$$\partial_t \beta(y, z, t) = -\mathcal{A}^\dagger \beta(y, z, t),$$

as derived in Appendix A.1.

**The Backward Distribution at Observation Time Points.**    Here, we calculate the backward distribution $\beta(y, z, t_i^-)$ right before the observation time points $\{t_i\}_{i \in 1,\ldots,N}$. We first note that

$$\beta(y, z, t_i - h) = p(x_i, \ldots, x_N \mid y, z, t_i - h)$$
$$= \frac{p(x_i, \ldots, x_N, y, z, t_i - h)}{p(y, z, t_i - h)}$$
$$= \frac{p(x_i \mid x_{i+1} \ldots, x_N, y, z, t_i - h)p(x_{i+1}, \ldots, x_N, y, z, t_i - h)}{p(y, z, t_i - h)}$$
$$= p(x_i \mid x_{i+1}, \ldots, x_N, y, z, t_i - h)p(x_{i+1}, \ldots, x_N \mid y, z, t_i - h).$$

Calculating $h \searrow 0$, we find

$$\beta(y, z, t_i^-) = \lim_{h \searrow 0} \beta(y, z, t_i - h) = p(x_i \mid y)\beta(y, z, t_i).$$

### A.2.3 Calculation of the Smoothing Distribution

We define the smoothing distribution as $\gamma(y, z, t) := p(y, z, t \mid x_{[1,N]}) = C^{-1}(t)\alpha(y, z, t)\beta(y, z, t)$. We find the dynamics of the smoothing distribution by calculating its time derivative. For this, we follow a proof analogous to [28]. By noting that $C^{-1}(t)$ is constant almost surely [A1], we obtain by differentiation

$$\partial_t p(y, z, t \mid x_{[1,N]}) = \partial_t \gamma(y, z, t) = \partial_t \left\{ C^{-1}(t)\alpha(y, z, t)\beta(y, z, t) \right\}$$
$$= C^{-1}(t)\alpha(y, z, t)\partial_t\beta(y, z, t) + C^{-1}(t)\beta(y, z, t)\partial_t\alpha(y, z, t). \tag{27}$$

The dynamics of the filtering distribution are

$$\partial_t \alpha(y, z, t) = -\sum_{i=1}^{n} \partial_{y_i} \left\{ f_i(y, z, t)\alpha(y, z, t) \right\} + \frac{1}{2} \sum_{i=1}^{n} \sum_{j=1}^{n} \partial_{y_i}\partial_{y_j} \left\{ D_{ij}(y, z, t)\alpha(y, z, t) \right\}$$
$$+ \sum_{z' \in \mathcal{Z}} \Lambda(z', z, t)\alpha(y, z', t),$$

where we define $\Lambda(z, z, t) := -\Lambda(z, t)$. The dynamics of the backward distribution are given as

$$\partial_t \beta(y, z, t) = -\sum_{i=1}^{n} f_i(y, z, t)\partial_{y_i}\beta(y, z, t) - \frac{1}{2} \sum_{i=1}^{n} \sum_{j=1}^{n} D_{ij}(y, z, t)\partial_{y_i}\partial_{y_j}\beta(y, z, t)$$
$$- \sum_{z' \in \mathcal{Z}} \Lambda(z, z', t)\beta(y, z', t).$$

Inserting the dynamics in Eq. (27) and using $C^{-1}(t)\alpha(y, z, t) = \frac{\gamma(y,z,t)}{\beta(y,z,t)}$ we find

$$\partial_t \gamma(y, z, t)$$
$$= \frac{\gamma(y, z, t)}{\beta(y, z, t)} \left( -\sum_{i=1}^{n} f_i(y, z, t)\partial_{y_i}\beta(y, z, t) - \frac{1}{2} \sum_{i=1}^{n} \sum_{j=1}^{n} D_{ij}(y, z, t)\partial_{y_i}\partial_{y_j}\beta(y, z, t) \right.$$
$$\left. - \sum_{z' \in \mathcal{Z}} \Lambda(z, z', t)\beta(y, z', t) \right)$$
$$+ \beta(y, z, t) \left( -\sum_{i=1}^{n} \partial_{y_i} \left\{ f_i(y, z, t)\frac{\gamma(y, z, t)}{\beta(y, z, t)} \right\} + \frac{1}{2} \sum_{i=1}^{n} \sum_{j=1}^{n} \partial_{y_i}\partial_{y_j} \left\{ D_{ij}(y, z, t)\frac{\gamma(y, z, t)}{\beta(y, z, t)} \right\} \right.$$
$$\left. + \sum_{z' \in \mathcal{Z}} \Lambda(z', z, t)\frac{\gamma(y, z', t)}{\beta(y, z', t)} \right). \tag{28}$$

Next we differentiate the intermediate terms using the product rule as

$$\partial_{y_i} \left\{ f_i(y, z, t)\frac{\gamma(y, z, t)}{\beta(y, z, t)} \right\}$$
$$= \frac{\left( \partial_{y_i} f_i(y, z, t)\gamma(y, z, t) + f_i(y, z, t)\partial_{y_i}\gamma(y, z, t) \right)\beta(y, z, t) - f_i(y, z, t)\gamma(y, z, t)\partial_{y_i}\beta(y, z, t)}{\beta(y, z, t)^2}$$
$$= \beta(y, z, t)^{-1} \left\{ \partial_{y_i} f_i(y, z, t)\gamma(y, z, t) + f_i(y, z, t)\partial_{y_i}\gamma(y, z, t) \right\}$$
$$- \beta(y, z, t)^{-2} f_i(y, z, t)\gamma(y, z, t)\partial_{y_i}\beta(y, z, t),$$

and

$$\partial_{y_i}\partial_{y_j}\left\{\frac{D_{ij}(y,z,t)\gamma(y,z,t)}{\beta(y,z,t)}\right\}$$

$$=\partial_{y_i}\left\{\gamma(y,z,t)\beta(y,z,t)^{-1}\partial_{y_j}D_{ij}(y,z,t)+D_{ij}(y,z,t)\beta(y,z,t)^{-1}\partial_{y_j}\gamma(y,z,t)\right.$$
$$\left.-D_{ij}(y,z,t)\gamma(y,z,t)\beta(y,z,t)^{-2}\partial_{y_j}\beta(y,z,t)\right\}$$

$$=\left(\gamma(y,z,t)\partial_{y_i}\partial_{y_j}D_{ij}(y,z,t)+\partial_{y_i}D_{ij}(y,z,t)\partial_{y_j}\gamma(y,z,t)\right)\beta(y,z,t)^{-1}$$
$$-\gamma(y,z,t)\beta(y,z,t)^{-2}\partial_{y_j}D_{ij}(y,z,t)\partial_{y_i}\beta(y,z,t)+\partial_{y_j}D_{ij}(y,z,t)\partial_{y_i}\gamma(y,z,t)\beta(y,z,t)^{-1}$$
$$+D_{ij}(y,z,t)\left(\beta(y,z,t)^{-1}\partial_{y_i}\partial_{y_j}\gamma(y,z,t)-\beta(y,z,t)^{-2}\partial_{y_j}\gamma(y,z,t)\partial_{y_i}\beta(y,z,t)\right)$$
$$-\partial_{y_i}D_{ij}(y,z,t)\gamma(y,z,t)\partial_{y_j}\beta(y,z,t)\beta(y,z,t)^{-2}$$
$$-D_{ij}(y,z,t)\left(\partial_{y_i}\gamma(y,z,t)\partial_{y_j}\beta(y,z,t)\beta(y,z,t)^{-2}\right.$$
$$\left.+\gamma(y,z,t)[\partial_{y_i}\partial_{y_j}\beta(y,z,t)\beta(y,z,t)^{-2}-2\partial_{y_j}\beta(y,z,t)\beta(y,z,t)^{-3}\partial_{y_i}\beta(y,z,t)]\right).$$

Collecting terms in $\beta(y,z,t)^{-1}$, $\beta(y,z,t)^{-2}$ and $\beta(y,z,t)^{-3}$, we find

$$\partial_{y_i}\partial_{y_j}\left\{\frac{D_{ij}(y,z,t)\gamma(y,z,t)}{\beta(y,z,t)}\right\}$$

$$=\beta(y,z,t)^{-1}\left\{\partial_{y_i}\partial_{y_j}D_{ij}(y,z,t)\gamma(y,z,t)+\partial_{y_j}D_{ij}(y,z,t)\partial_{y_i}\gamma(y,z,t)\right.$$
$$\left.+\partial_{y_i}D_{ij}(y,z,t)\partial_{y_j}\gamma(y,z,t)+D_{ij}(y,z,t)\partial_{y_i}\partial_{y_j}\gamma(y,z,t)\right\}$$
$$-\beta(y,z,t)^{-2}\left\{\partial_{y_j}D_{ij}(y,z,t)\gamma(y,z,t)\partial_{y_i}\beta(y,z,t)\right.$$
$$+\partial_{y_i}D_{ij}(y,z,t)\gamma(y,z,t)\partial_{y_j}\beta(y,z,t)$$
$$+D_{ij}(y,z,t)\partial_{y_j}\gamma(y,z,t)\partial_{y_i}\beta(y,z,t)$$
$$+D_{ij}(y,z,t)\partial_{y_i}\gamma(y,z,t)\partial_{y_j}\beta(y,z,t)$$
$$\left.+D_{ij}(y,z,t)\gamma(y,z,t)\partial_{y_i}\partial_{y_j}\beta(y,z,t)\right\}$$
$$+\beta(y,z,t)^{-3}\left\{2D_{ij}(y,z,t)\gamma(y,z,t)\partial_{y_i}\beta(y,z,t)\partial_{y_j}\beta(y,z,t)\right\}.$$

Using the terms in Eq. (28) we have

$$\partial_t\gamma(y,z,t)$$

$$=-\sum_{i=1}^{n}f_i(y,z,t)\gamma(y,z,t)\beta(y,z,t)^{-1}\partial_{y_i}\beta(y,z,t)$$

$$-\frac{1}{2}\sum_{i=1}^{n}\sum_{j=1}^{n}D_{ij}(y,z,t)\gamma(y,z,t)\beta(y,z,t)^{-1}\partial_{y_i}\partial_{y_j}\beta(y,z,t)$$

$$-\sum_{i=1}^{n}\beta(y,z,t)^{-1}\left\{[\partial_{y_i}f_i(y,z,t)\gamma(y,z,t)+f_i(y,z,t)\partial_{y_i}\gamma(y,z,t)]\beta(y,z,t)\right.$$
$$\left.-f_i(y,z,t)\gamma(y,z,t)\partial_{y_i}\beta(y,z,t)\right\}$$

$$+\frac{1}{2}\sum_{i=1}^{n}\sum_{j=1}^{n}\left[\partial_{y_i}\partial_{y_j}D_{ij}(y,z,t)\gamma(y,z,t)+\partial_{y_j}D_{ij}(y,z,t)\partial_{y_i}\gamma(y,z,t)\right.$$

$$+\partial_{y_i}D_{ij}(y,z,t)\partial_{y_j}\gamma(y,z,t)+D_{ij}(y,z,t)\partial_{y_i}\partial_{y_j}\gamma(y,z,t)$$
$$-\beta(y,z,t)^{-1}\left\{\partial_{y_j}D_{ij}(y,z,t)\gamma(y,z,t)\partial_{y_i}\beta(y,z,t)\right.$$
$$+\partial_{y_i}D_{ij}(y,z,t)\gamma(y,z,t)\partial_{y_j}\beta(y,z,t)$$
$$+D_{ij}(y,z,t)\partial_{y_j}\gamma(y,z,t)\partial_{y_i}\beta(y,z,t)$$
$$+D_{ij}(y,z,t)\partial_{y_i}\gamma(y,z,t)\partial_{y_j}\beta(y,z,t)$$
$$\left.+D_{ij}(y,z,t)\gamma(y,z,t)\partial_{y_i}\partial_{y_j}\beta(y,z,t)\right\}$$
$$\left.+\beta(y,z,t)^{-2}\left\{2D_{ij}(y,z,t)\gamma(y,z,t)\partial_{y_i}\beta(y,z,t)\partial_{y_j}\beta(y,z,t)\right\}\right]$$

$$+\sum_{z'\in\mathcal{Z}}\left(\Lambda(z',z,t)\frac{\beta(y,z,t)}{\beta(y,z',t)}\gamma(y,z',t)-\Lambda(z,z',t)\frac{\beta(y,z',t)}{\beta(y,z,t)}\gamma(y,z,t)\right)$$

$$
\begin{aligned}
= & -\sum_{i=1}^{n} \partial_{y_i} f_i(y,z,t)\gamma(y,z,t) + f_i(y,z,t)\partial_{y_i}\gamma(y,z,t) \\
& + \frac{1}{2}\sum_{i=1}^{n}\sum_{j=1}^{n} \partial_{y_i}\partial_{y_j} D_{ij}(y,z,t)\gamma(y,z,t) + \partial_{y_j}D_{ij}(y,z,t)\partial_{y_i}\gamma(y,z,t) \\
& \qquad\qquad + \partial_{y_i}D_{ij}(y,z,t)\partial_{y_j}\gamma(y,z,t) + D_{ij}(y,z,t)\partial_{y_i}\partial_{y_j}\gamma(y,z,t) \\
& - \frac{\beta(y,z,t)^{-1}}{2}\sum_{i=1}^{n}\sum_{j=1}^{n}\big\{\partial_{y_j}D_{ij}(y,z,t)\gamma(y,z,t)\partial_{y_i}\beta(y,z,t) \\
& \qquad\qquad + \partial_{y_i}D_{ij}(y,z,t)\gamma(y,z,t)\partial_{y_j}\beta(y,z,t) \\
& \qquad\qquad + 2D_{ij}(y,z,t)\gamma(y,z,t)\partial_{y_i}\partial_{y_j}\beta(y,z,t) \\
& \qquad\qquad + D_{ij}(y,z,t)\partial_{y_j}\gamma(y,z,t)\partial_{y_i}\beta(y,z,t) \\
& \qquad\qquad + D_{ij}(y,z,t)\partial_{y_i}\gamma(y,z,t)\partial_{y_j}\beta(y,z,t)\big\} \\
& + \beta(y,z,t)^{-2}\sum_{i=1}^{n}\sum_{j=1}^{n} D_{ij}(y,z,t)\gamma(y,z,t)\partial_{y_i}\beta(y,z,t)\partial_{y_j}\beta(y,z,t) \\
& + \sum_{z'\in\mathcal{Z}\backslash z}\left(\Lambda(z',z,t)\frac{\beta(y,z,t)}{\beta(y,z',t)}\gamma(y,z',t) - \Lambda(z,z',t)\frac{\beta(y,z',t)}{\beta(y,z,t)}\gamma(y,z,t)\right) \\
= & -\sum_{i=1}^{n}\big\{\partial_{y_i}f_i(y,z,t)\gamma(y,z,t) + f_i(y,z,t)\partial_{y_i}\gamma(y,z,t) \\
& \qquad + \frac{\beta(y,z,t)^{-1}}{2}\sum_{j=1}^{n}\big[\partial_{y_j}D_{ij}(y,z,t)\gamma(y,z,t)\partial_{y_i}\beta(y,z,t) \\
& \qquad\qquad + \partial_{y_i}D_{ij}(y,z,t)\gamma(y,z,t)\partial_{y_j}\beta(y,z,t) \\
& \qquad\qquad + 2D_{ij}(y,z,t)\gamma(y,z,t)\partial_{y_i}\partial_{y_j}\beta(y,z,t) \\
& \qquad\qquad + D_{ij}(y,z,t)\partial_{y_j}\gamma(y,z,t)\partial_{y_i}\beta(y,z,t) \\
& \qquad\qquad + D_{ij}(y,z,t)\partial_{y_i}\gamma(y,z,t)\partial_{y_j}\beta(y,z,t)\big] \\
& \qquad -\beta(y,z,t)^{-2}\sum_{j=1}^{n} D_{ij}(y,z,t)\gamma(y,z,t)\partial_{y_i}\beta(y,z,t)\partial_{y_j}\beta(y,z,t)\big\} \\
& + \frac{1}{2}\sum_{i=1}^{n}\sum_{j=1}^{n}\partial_{y_i}\big\{\partial_{y_j}D_{ij}(y,z,t)\gamma(y,z,t) + D_{ij}(y,z,t)\partial_{y_j}\gamma(y,z,t)\big\} \\
& + \sum_{z'\in\mathcal{Z}}\Lambda(z',z,t)\frac{\beta(y,z,t)}{\beta(y,z',t)}\gamma(y,z',t) \\
= & -\sum_{i=1}^{n}\big\{\partial_{y_i}\{f_i(y,z,t)\gamma(y,z,t)\} \\
& \qquad + \beta(y,z,t)^{-1}\sum_{j=1}^{n}\big[\partial_{y_i}D_{ij}(y,z,t)\gamma(y,z,t)\partial_{y_j}\beta(y,z,t) \\
& \qquad\qquad + D_{ij}(y,z,t)\gamma(y,z,t)\partial_{y_i}\partial_{y_j}\beta(y,z,t) \\
& \qquad\qquad + D_{ij}(y,z,t)\partial_{y_i}\gamma(y,z,t)\partial_{y_j}\beta(y,z,t)\big] \\
& \qquad -\beta(y,z,t)^{-2}\sum_{j=1}^{n} D_{ij}(y,z,t)\gamma(y,z,t)\partial_{y_i}\beta(y,z,t)\partial_{y_j}\beta(y,z,t)\big\} \\
& + \frac{1}{2}\sum_{i=1}^{n}\sum_{j=1}^{n}\partial_{y_i}\partial_{y_j}\{D_{ij}(y,z,t)\gamma(y,z,t)\} + \sum_{z'\in\mathcal{Z}}\Lambda(z',z,t)\frac{\beta(y,z,t)}{\beta(y,z',t)}\gamma(y,z',t).
\end{aligned}
$$

Next, collecting terms by making use of the product rule, we have

$$\partial_t \gamma(y, z, t)$$

$$= -\sum_{i=1}^{n} \partial_{y_i} \left\{ f_i(y, z, t)\gamma(y, z, t) + \sum_{j=1}^{n} D_{ij}(y, z, t)\partial_{y_j}\beta(y, z, t)\beta(y, z, t)^{-1}\gamma(y, z, t) \right\}$$

$$+ \frac{1}{2}\sum_{i=1}^{n}\sum_{j=1}^{n} \partial_{y_i}\partial_{y_j} \left\{ D_{ij}(y, z, t)\gamma(y, z, t) \right\} + \sum_{z' \in \mathcal{Z}} \Lambda(z', z, t)\frac{\beta(y, z, t)}{\beta(y, z', t)}\gamma(y, z', t)$$

$$= -\sum_{i=1}^{n} \partial_{y_i} \left\{ \left( f_i(y, z, t) + \sum_{j=1}^{n} D_{ij}(y, z, t)\partial_{y_j}\log\beta(y, z, t) \right) \gamma(y, z, t) \right\}$$

$$+ \frac{1}{2}\sum_{i=1}^{n}\sum_{j=1}^{n} \partial_{y_i}\partial_{y_j} \left\{ D_{ij}(y, z, t)\gamma(y, z, t) \right\} + \sum_{z' \in \mathcal{Z}} \Lambda(z', z, t)\frac{\beta(y, z, t)}{\beta(y, z', t)}\gamma(y, z', t).$$

Finally, we can write

$$\partial_t p(y, z, t \mid x_{[1,N]}) = \tilde{\mathcal{A}}p(y, z, t \mid x_{[1,N]})$$

$$= -\sum_{i=1}^{n} \partial_{y_i} \left\{ \tilde{f}_i(y, z, t)p(y, z, t \mid x_{[1,N]}) \right\}$$

$$+ \frac{1}{2}\sum_{i=1}^{n}\sum_{j=1}^{n} \partial_{y_i}\partial_{y_j} \left\{ \tilde{D}_{ij}(y, z, t)p(y, z, t \mid x_{[1,N]}) \right\}$$

$$+ \sum_{z' \in \mathcal{Z}} \tilde{\Lambda}(z', z, t)p(y, z', t \mid x_{[1,N]}),$$

with the posterior drift

$$\tilde{f}_i(y, z, t) = f_i(y, z, t) + \sum_{j=1}^{n} D_{ij}(y, z, t)\partial_{y_j}\log\beta(y, z, t),$$

the posterior dispersion

$$\tilde{D}_{ij}(y, z, t) = D_{ij}(y, z, t),$$

and the posterior rate

$$\tilde{\Lambda}(z', z, t) = \Lambda(z', z, t)\frac{\beta(y, z, t)}{\beta(y, z', t)}.$$

### A.3  Approximate Inference

#### A.3.1  The Path-Wise Kullback-Leibler Divergence Between Hybrid Processes

To derive the KL divergence $\mathrm{KL}\left(\mathbb{Q}_{Y,Z} \mid\mid \mathbb{P}_{Y,Z}\right)$ between two hybrid processes $\{Y^Q(t), Z^Q(t)\}_{t \geq 0}$, $\{Y^P(t), Z^P(t)\}_{t \geq 0}$ with the respective path measures $\mathbb{Q}_{Y,Z}, \mathbb{P}_{Y,Z}$, we consider discretized versions of the continuous processes on a regular time grid, $t \in \{0, h, 2h, ..., K \cdot h = T\}$ for some small $h$, and aim to take the limit $h \to 0$ of the resulting expressions.

For the discretized joint paths $\{Y_k, Z_k\}_{k \in \{0,1,...,K\}}$, we can explicitly write down the probability density functions; we abbreviate $\mathbf{y} := \{y_k\}_{k \in \{0,1,...,K\}}$ and $\mathbf{z} := \{z_k\}_{k \in \{0,1,...,K\}}$ and write

$$q(\mathbf{y}, \mathbf{z}) = q(y_0, z_0)\prod_{k=1}^{K} q(y_k, z_k \mid y_{k-1}, z_{k-1}),$$

$$p(\mathbf{y}, \mathbf{z}) = p(y_0, z_0)\prod_{k=1}^{K} p(y_k, z_k \mid y_{k-1}, z_{k-1}).$$

Inserting both into the KL divergence yields

$$
\mathrm{KL}\left(\mathbb{Q}_{Y,Z} \mid\mid \mathbb{P}_{Y,Z}\right) = \sum_{z_0,\ldots,z_K} \int q(y_0,z_0) \prod_{k=1}^{K} q(y_k,z_k \mid y_{k-1},z_{k-1})
$$

$$
\left( \ln \frac{q(y_0,z_0)}{p(z_0,y_0)} + \sum_{j=1}^{K} \ln \frac{q(y_k,z_k \mid y_{k-1},z_{k-1})}{p(y_k,z_k \mid y_{k-1},z_{k-1})} \right) \mathrm{d}y_0 \cdots \mathrm{d}y_K
$$

$$
= \mathrm{KL}\left(\mathbb{Q}_{Y,Z}^0 \mid\mid \mathbb{P}_{Y,Z}^0\right) + \sum_{j=1}^{K} \sum_{z_0,\ldots,z_K} \int \left( q(y_0,z_0) \prod_{i=1}^{j} q(y_k,z_k \mid y_{k-1},z_{k-1}) \right.
$$

$$
\left. \ln \frac{q(y_k,z_k \mid y_{k-1},z_{k-1})}{p(y_k,z_k \mid y_{k-1},z_{k-1})} \right) \mathrm{d}y_0 \cdots \mathrm{d}y_K, \quad (29)
$$

where we introduced $\mathrm{KL}\left(\mathbb{Q}_{Y,Z}^0 \mid\mid \mathbb{P}_{Y,Z}^0\right)$ for the initial distributions: as detailed in the main paper, we assume $q(y,z,0) = q(z,0)q(y,0 \mid z)$ and $p(y,z,0) = p(z,0)p(y,0 \mid z)$, which yields

$$
\mathrm{KL}\left(\mathbb{Q}_{Y,Z}^0 \mid\mid \mathbb{P}_{Y,Z}^0\right) = \sum_z \int q(z,0)q(y,0 \mid z) \ln \frac{q(z,0)q(y,0 \mid z)}{p(z,0)p(y,0 \mid z)} \, \mathrm{d}y
$$

$$
= \sum_z \int q(z,0)q(y,0;z) \ln \frac{q(y,0 \mid z)}{p(y,0 \mid z)} \, \mathrm{d}y + \sum_z q(z,0) \ln \frac{q(z,0)}{p(z,0)}
$$

$$
= \sum_z q(z,0) \, \mathrm{KL}\left(\mathbb{Q}_{Y|Z}^0 \mid\mid \mathbb{P}_{Y|Z}^0\right) + \mathrm{KL}\left(\mathbb{Q}_Z^0 \mid\mid \mathbb{P}_Z^0\right). \quad (30)
$$

Any of the $K$ summands from the second term of Eq. (29) can be simplified as

$$
\sum_{z_0,\ldots,z_K} \int \left( q(y_0,z_0) \prod_{i=1}^{j} q(y_k,z_k \mid y_{k-1},z_{k-1}) \right) \ln \frac{q(y_k,z_k \mid y_{k-1},z_{k-1})}{p(y_k,z_k \mid y_{k-1},z_{k-1})} \mathrm{d}y_0 \cdots \mathrm{d}y_K
$$

$$
= \sum_{z_{k-1},z_k} \int q(y_k,z_k \mid y_{k-1},z_{k-1})q(y_{k-1},z_{k-1}) \ln \frac{q(y_k,z_k \mid y_{k-1},z_{k-1})}{p(y_k,z_k \mid y_{k-1},z_{k-1})} \mathrm{d}y_{k-1}\mathrm{d}y_k
$$

$$
= \sum_{z_{k-1},z_k} \int q(y_k,z_k \mid y_{k-1},z_{k-1})q(y_{k-1},z_{k-1}) \ln \frac{q(y_k \mid y_{k-1},z_{k-1})}{p(y_k \mid y_{k-1},z_{k-1})} \frac{q(z_k \mid z_{k-1})}{p(z_k \mid z_{k-1})} \mathrm{d}y_{k-1}\mathrm{d}y_k
$$

$$
= \sum_{z_{k-1},z_k} \int q(y_k,z_k \mid y_{k-1},z_{k-1})q(y_{k-1},z_{k-1}) \ln \frac{q(y_k \mid y_{k-1},z_k,z_{k-1})}{p(y_k \mid y_{k-1},z_k,z_{k-1})} \mathrm{d}y_{k-1}\mathrm{d}y_k
$$

$$
+ \sum_{z_{k-1},z_k} q(z_k,z_{k-1}) \ln \frac{q(z_k \mid z_{k-1})}{p(z_k \mid z_{k-1})}. \quad (31)
$$

Inspecting

$$
q(y_k,z_k \mid y_{k-1},z_{k-1}) = q(y_k \mid z_k,y_{k-1},z_{k-1})q(z_k \mid z_{k-1}),
$$

and expanding $q(z_k \mid z_{k-1})$, we find

$$
q(y_z,z_k \mid y_{k-1},z_{k-1}) = \begin{cases} q(y_k \mid z_{k-1},y_{k-1}) & \text{if } z_k = z_{k-1} \\ q(y_k \mid z_k,z_{k-1},y_{k-1})\Lambda_{z_{k-1},z_k}h + o\left(h\right), & \text{otherwise.} \end{cases}
$$

Recalling Eqs. (22) and (23), we furthermore have

$$
q(y_k \mid z_k,z_{k-1},y_{k-1}) = \delta(y_k - y_{k-1}) + o\left(1\right),
$$

which can intuitively be understood as the process being continuous, thinking of $\delta(y_k - y_{k-1})$ as point-masses located at $y_k = y_{k-1}$, and hence, keeping only terms linear in $h$:

$$
q(y_z,z_k \mid y_{k-1},z_{k-1}) = \begin{cases} q(y_k \mid z_{k-1},y_{k-1}) & \text{if } z_k = z_{k-1} \\ \delta(y_k - y_{k-1})\Lambda_{z_{k-1},z_k}h + o\left(h\right), & \text{otherwise.} \end{cases}
$$

This yields

$$\sum_{z_0,\dots,z_K} \int \left( q(y_0,z_0) \prod_{i=1}^{j} q(y_k,z_k \mid y_{k-1},z_{k-1}) \right) \ln \frac{q(y_k,z_k \mid y_{k-1},z_{k-1})}{p(y_k,z_k \mid y_{k-1},z_{k-1})} \mathrm{d}y_0 \cdots \mathrm{d}y_K$$

$$= \sum_{z_{k-1}} \int q(y_k \mid z_{k-1},y_{k-1}) q(y_{k-1},z_{k-1}) \ln \frac{q(y_k \mid y_{k-1},z_{k-1})}{p(y_k \mid y_{k-1},z_{k-1})} \mathrm{d}y_{k-1}\mathrm{d}y_k$$

$$+ \sum_{\substack{z_{k-1},z_k \\ z_{k-1}\neq z_k}} \int \left( \delta(y_k - y_{k-1})\Lambda_{z_{k-1},z_k} h + o\left(h\right) \right) q(y_{k-1},z_{k-1})$$

$$\cdot \ln \frac{q(y_k \mid y_{k-1},z_k,z_{k-1})}{p(y_k \mid y_{k-1},z_k,z_{k-1})} \mathrm{d}y_{k-1}\mathrm{d}y_k$$

Because of $\delta(y_k - y_{k-1})$, the integrand will only contribute for $y_k = y_{k-1}$; however, we also have

$$\ln \frac{q(y_k \mid y_{k-1},z_k,z_{k-1})}{p(y_k \mid y_{k-1},z_k,z_{k-1})} = \ln \frac{\delta(y_k - y_{k-1}) + o(1)}{\delta(y_k - y_{k-1}) + o(1)} \xrightarrow{h\to 0} 0.$$

Inspecting the other log-fraction, we find

$$\frac{q(y_k \mid y_{k-1},z_{k-1})}{p(y_k \mid y_{k-1},z_{k-1})} = \frac{\exp\{-\frac{1}{2h}\|y_k - y_{k-1} - g(y_{k-1},z_{k-1},(k-1)h)\cdot h\|_{D^{-1}}^2\}}{\exp\{-\frac{1}{2h}\|y_k - y_{k-1} - f(y_{k-1},z_{k-1},(k-1)h)\cdot h\|_{D^{-1}}^2\}}$$

$$= \exp\left\{ \frac{h}{2} \cdot \|g(y_{k-1},z_{k-1},(k-1)h) - f(y_{k-1},z_{k-1},(k-1)h)\|_{D^{-1}}^2 \right\}$$

and hence the ratio does not depend on the time step $k$, but only $k-1$. We hence can integrate out $z_k, y_k$. Note that the normalizing prefactor $(2\pi)^{\frac{k}{2}}|D|^{-\frac{1}{2}}$ is the same for both distributions and hence cancels out; if the processes $\mathbb{Q}$ and $\mathbb{P}$ had different dispersions $D_{\mathbb{Q}}$ and $D_{\mathbb{P}}$, this cancellation would not occur and the KL would diverge [67].

Taking the limit $K \to \infty$, $h \to 0$ with $K \cdot h = T$ yields an integral expression:

$$\int_0^T \sum_z \int q(y,z,t)\frac{1}{2}\|g(y,z,t) - f(y,z,t)\|_{D^{-1}}^2 \, \mathrm{d}y\mathrm{d}t$$

$$= \frac{1}{2}\int_0^T \mathsf{E}\left[\|(g(y,z,t) - f(y,z,t)\|_{D^{-1}}^2\right] \mathrm{d}t.$$

The second term of Eq. (31) does not depend on the $Y$-process and hence is simply the KL divergence between two MJPs. Its derivation is completely analogous to the one presented above, which is why we omit the details and refer the interested reader to, e.g., [29]. The KL divergence is found to be

$$\int_0^T \sum_z q_Z(z,t) \left[ \sum_{z'\in\mathcal{Z}\setminus z} \left\{ \tilde{\Lambda}(z,z',t) \left( \ln\tilde{\Lambda}(z,z',t) - \ln\Lambda(z,z',t) \right) \right\} - (\tilde{\Lambda}(z,t) - \Lambda(z,t)) \right] \mathrm{d}t$$

$$= \int_0^T \mathsf{E}\left[ \sum_{z'\in\mathcal{Z}\setminus z} \left\{ \tilde{\Lambda}(z,z',t) \left( \ln\tilde{\Lambda}(z,z',t) - \ln\Lambda(z,z',t) \right) \right\} - (\tilde{\Lambda}(z,t) - \Lambda(z,t)) \right] \qquad (32)$$

with the variational rates $\tilde{\Lambda}$ and the prior rates $\Lambda$ as defined in the main text. The full KL divergence finally reads

$$\mathrm{KL}\left(\mathbb{Q}_{Y,Z} \,\|\, \mathbb{P}_{Y,Z}\right) = \mathrm{KL}\left(\mathbb{Q}_{Y,Z}^0 \,\|\, \mathbb{P}_{Y,Z}^0\right) + \frac{1}{2}\int_0^T \mathsf{E}\left[\|(g(y,z,t) - f(y,z,t)\|_{D^{-1}}^2\right.$$

$$\left. + \sum_{z'\in\mathcal{Z}\setminus z} \left\{ \tilde{\Lambda}(z,z',t) \left( \ln\tilde{\Lambda}(z,z',t) - \ln\Lambda(z,z',t) \right) \right\} - (\tilde{\Lambda}(z,t) - \Lambda(z,t)) \right] \mathrm{d}t. \qquad (33)$$

### A.3.2 Derivation of the Constrained Variational Dynamics

The separate dynamic constraints on the variational factors $q_Z(z,t)$, $q_Y(y,t \mid z)$ do not in general ensure the HME to be fulfilled. For meta-stable systems, however, this is a reasonable approximation; to see this, consider the HME

$$\partial_t q(y,z,t) = -\sum_{i=1}^{n} \partial_{y_i} \{g_i(y,z,t)q(y,z,t)\}$$
$$+ \frac{1}{2}\sum_{i=1}^{n}\sum_{j=1}^{n} \partial_{y_i}\partial_{y_j}\{D_{ij}q(y,z,t)\} + \sum_{z' \in \mathcal{Z}} \tilde{\Lambda}(z',z,t)q(y,z',t),$$

and insert the structured mean-field approximation $q(y,z,t) = q_Z(z,t)q_Y(y,t \mid z)$:

$$\partial_t q(y,z,t) = \partial_t\{q_Z(z,t)q_Y(y,t \mid z)\}$$
$$= q_Y(y,t \mid z)\partial_t q_Z(z,t) + q_Z(z,t)\partial_t q_Y(y,t \mid z)$$
$$= q_Z(z,t)\left(-\sum_{i=1}^{n}\partial_{y_i}\{g_i(y,z,t)q_Y(y,t \mid z)\} + \frac{1}{2}\sum_{i=1}^{n}\sum_{j=1}^{n}\partial_{y_i}\partial_{y_j}\{D_{ij}q_Y(y,t \mid z)\}\right)$$
$$+ \sum_{z' \in \mathcal{Z}\setminus z}\tilde{\Lambda}(z',z,t)q_Z(z',t)q_Y(y,t \mid z') - \tilde{\Lambda}(z,t)q_Z(z,t)q_Y(y,t \mid z). \tag{34}$$

Collecting terms, we find

$$q_Z(z,t)\left(\partial_t q_Y(y,t \mid z) + \sum_{i=1}^{n}\partial_{y_i}\{g_i(y,z,t)q_Y(y,t \mid z)\} - \frac{1}{2}\sum_{i=1}^{n}\sum_{j=1}^{n}\partial_{y_i}\partial_{y_j}\{D_{ij}q_Y(y,t \mid z)\}\right)$$
$$= -q_Y(y,t \mid z)\partial_t q_Z(z,t) + \sum_{z' \in \mathcal{Z}\setminus z}\tilde{\Lambda}(z',z,t)q_Z(z',t)q_Y(y,t \mid z') - \tilde{\Lambda}(z,t)q_Z(z,t)q_Y(y,t \mid z). \tag{35}$$

If we are almost certain to be in state $z$ at time $t$, $q(z,t) \approx 1$, and the exit rate $\tilde{\Lambda}(z,t)$ from this state is small, i.e., the remain time is large and the state is meta-stable, we have

$$-q_Y(y,t \mid z)\partial_t q_Z(z,t) + \sum_{z' \in \mathcal{Z}\setminus z}\tilde{\Lambda}(z',z,t)q_Z(z',t)q_Y(y,t \mid z') - \tilde{\Lambda}(z,t)q_Z(z,t)q_Y(y,t \mid z)$$
$$\approx -q_Y(y,t \mid z)\partial_z q_Z(z,t) + \tilde{\Lambda}(z,t)q_Z(z,t)q_Y(y,t \mid z)$$
$$= q_Y(y,t \mid z)\underbrace{\left(-\partial_t q_Z(z,t) + \tilde{\Lambda}(z,t)q_Z(z,t)\right)}_{\approx 0}, \tag{36}$$

since we know that the master equation holds (see above). Accordingly, both sides of Eq. (35) have to vanish, that is, $q_Z(z,t)$ and $q_Y(y,t \mid z)$ have to individually follow the master equation and the FPE. The higher the uncertainty in the mode assignment $q(z,t)$, the larger the approximation error; we expect the approximation to be of high quality in regions where $z$ does not change rapidly. This is acceptable, since we are genuinely interested in meta-stable systems, which by definition only sparingly transition between distinct, qualitatively different modes.

### A.3.3 Computing the Optimal Variational Distribution

We restate the ELBO $\mathcal{L}$ Eq. (10) as well as the full Lagrangian $L$ Eq. (16) utilizing the shorthand notation $g - f := g(y,z,t) - f(y,z,t)$:

$$\mathcal{L} = \int_0^T \ell_{\mathbb{Q}}(t)\,\mathrm{d}t, \tag{37}$$

with

$$\ell_{\mathbb{Q}}(t) = - \mathsf{E} \left[ \frac{1}{2} \|(g - f)\|_{D^{-1}}^2 + \sum_{z' \in \mathcal{Z} \setminus z} \left\{ \tilde{\Lambda}(z, z', t) \ln \frac{\tilde{\Lambda}(z, z', t)}{\Lambda(z, z', t)} \right\} - (\tilde{\Lambda}(z, t) - \Lambda(z, t)) \right]$$

$$- \mathsf{E} \left[ \ln \frac{q(y, z, 0)}{p(y, z, 0)} \right] \delta(t - 0) + \sum_{i=1}^{N} \mathsf{E}_{\mathbb{Q}_{Y,Z}} \left[ \ln p(x_i \mid y_i) \right] \delta(t - t_i),$$

(38)

and

$$L = \int_0^T \ell(t) \, \mathrm{d}t,$$

(39)

where

$$\ell(t) = \ell_{\mathbb{Q}} + \sum_{z \in \mathcal{Z}} \left[ \lambda^\top(z, t) \left( \dot{\mu}(z, t) - (A(z, t)\mu(z, t) + b(z, t)) \right) \right.$$

(40)

$$+ \operatorname{tr} \left\{ \Psi^\top(z, t) \left( \dot{\Sigma}(z, t) - \left( A(z, t)\Sigma(z, t) + \Sigma(z, t)A^\top(z, t) + D \right) \right) \right\}$$

$$+ \nu(z, t) \left( \dot{q}_Z(z, t) - \sum_{z' \in \mathcal{Z}} \tilde{\Lambda}_{z'z}(t) q_Z(z', t) \right) \right].$$

**Stationarity with respect to the Variational MJP**   The EL equation for $q_Z$ reads

$$\frac{\mathrm{d}}{\mathrm{d}t} \frac{\partial \ell}{\partial \dot{q}_Z} = \frac{\partial \ell}{\partial q_Z}.$$

With Eq. (40), we therefore find the components

$$\frac{\mathrm{d}}{\mathrm{d}t} \frac{\partial \ell}{\partial \dot{q}_Z(z, t)} = \frac{\mathrm{d}}{\mathrm{d}t} \nu(z, t)$$

and

$$\frac{\partial \ell}{\partial q_Z(z, t)} = \frac{\partial \ell_{\mathbb{Q}}}{\partial q_Z(z, t)} - \sum_{z' \in \mathcal{Z} \setminus z} \tilde{\Lambda}(z, z', t)\nu(z', t) + \tilde{\Lambda}(z, t)\nu(z, t).$$

Hence, the EL equation yields

$$\frac{\mathrm{d}}{\mathrm{d}t} \nu(z, t) = \frac{\partial \ell_{\mathbb{Q}}}{\partial q_Z(z, t)} - \sum_{z' \in \mathcal{Z} \setminus z} \tilde{\Lambda}(z, z', t)\nu(z', t) + \tilde{\Lambda}(z, t)\nu(z, t).$$

(41)

Using the law of total expectation

$$\mathsf{E}\left[ \phi(Y(t), Z(t), t) \right] = \sum_z q_Z(z, t) \, \mathsf{E}\left[ \phi(Y(t), Z(t), t) \mid Z(t) = z \right],$$

we calculate the gradient of Eq. (38) with respect to $q_Z(z, t)$ as

$$\partial_{q_Z(z,t)} \ell_{\mathbb{Q}} = - \mathsf{E}\left[ \|g - f\|_{D^{-1}}^2 | z \right]$$

$$- \sum_{z' \in \mathcal{Z} \setminus z} \left[ \tilde{\Lambda}_{zz'}(t) \left( \ln \frac{\tilde{\Lambda}_{zz'}(t)}{\Lambda_{zz'}} - 1 + \nu(z', t) - \nu(z, t) \right) + \Lambda_{zz'} \right]$$

$$+ \sum_{i}^{N} \mathsf{E}\left[ \ln p(x_i \mid y_i) | z \right] \delta(t - t_i).$$

For linear prior drift functions $f(y, z, t) = A_p(z, t)y + b_p(z, t)$, we can calculate $\mathsf{E}\left[ \|g - f\|_{D^{-1}}^2 | z \right]$ explicitly. We use $\bar{A}(z, t) := A(z, t) - A_p(z, t)$ and $\bar{b}(z, t) := b(z, t) - b_p(z, t)$ and obtain

$$\mathsf{E}\left[ \|g - f\|_{D^{-1}}^2 | z \right] = \operatorname{tr} \left\{ \bar{A}(z, t)^\top D^{-1} \bar{A}(z, t)\Sigma(z, t) \right\}$$

$$+ (\bar{A}(z, t)\mu(z, t) + \bar{b}(z, t))^\top D^{-1} (\bar{A}(z, t)\mu(z, t) + \bar{b}(z, t)).$$

Therefore, the gradient $\partial_{q_Z(z,t)}\ell_\mathbb{Q}$ reads

$$
\begin{aligned}
\partial_{q_Z(z,t)}\ell_\mathbb{Q} = & -\operatorname{tr}\left\{\bar{A}(z,t)^\top D^{-1}\bar{A}(z,t)\Sigma(z,t)\right\} \\
& -\left(\bar{A}(z,t)\mu(z,t) + \bar{b}(z,t)\right)^\top D^{-1}\left(\bar{A}(z,t)\mu(z,t) + \bar{b}(z,t)\right) \\
& -\sum_{z'\in\mathcal{Z}\backslash z}\left[\tilde{\Lambda}_{zz'}(t)\left(\ln\frac{\tilde{\Lambda}_{zz'}(t)}{\Lambda_{zz'}} - 1 + \nu(z',t) - \nu(z,t)\right) + \Lambda_{zz'}\right] \\
& +\sum_{i=1}^N \mathsf{E}\left[\ln p(x_i\mid y_i)\mid z\right]\delta(t - t_i).
\end{aligned}
$$

Due to the delta-contributions $\delta(t - t_i)$, the evolution equation for the Lagrange multiplier $\nu(z,t)$ Eq. (17) is an impulsive differential equation [50] which can be solved piece-wise by integrating the ODE between the discontinuities (starting at $\nu(z,T) = 0$) and applying reset conditions at the integration boundaries, similar to exact posterior inference (c.f. Appendix A.2.3):

$$
\nu(z,t_i) = \mathsf{E}\left[\ln p(x_i\mid y_i)\mid z\right] + \nu(z,t_i^-), \tag{42}
$$

with $\nu(z,t_i^-) := \lim_{h\searrow 0}\nu(z,t_i - h)$.

For Gaussian observation noise, we have

$$
\begin{aligned}
\mathsf{E}\left[\ln p(x_i\mid y_i)\mid z\right] = -\frac{1}{2}\Big\{ & n\ln(2\pi) + \ln|\Sigma_{\mathrm{obs}}| \\
& + (x_i - \mu(z,t_i))^\top\Sigma_{\mathrm{obs}}^{-1}(x_i - \mu(z,t_i)) + \operatorname{tr}\left\{\Sigma_{\mathrm{obs}}^{-1}\Sigma(z,t_i)\right\}\Big\}. \tag{43}
\end{aligned}
$$

**Stationarity with respect to the Variational GPs** In the same manner as for the variational MJP, we straightforwardly arrive at

$$
\begin{aligned}
\frac{\mathrm{d}}{\mathrm{d}t}\lambda(z,t) &= \partial_{\mu(z,t)}\ell_\mathbb{Q} - A^\top(z,t)\lambda(z,t), \\
\frac{\mathrm{d}}{\mathrm{d}t}\Psi(z,t) &= \partial_{\Sigma(z,t)}\ell_\mathbb{Q} - A^\top(z,t)\Psi(z,t) - \Psi(z,t)A(z,t).
\end{aligned} \tag{44}
$$

We find the gradients as

$$
\partial_{\mu(z,t)}\ell_\mathbb{Q} = -\partial_{\mu(z,t)}\mathsf{E}\left[\|g - f\|_{D^{-1}}^2\right] + \sum_{i=1}^N \partial_{\mu(z,t)}\mathsf{E}\left[\ln p(x_i\mid y_i)\right]\delta(t - t_i),
$$

$$
\partial_{\Sigma(z,t)}\ell_\mathbb{Q} = -\partial_{\Sigma(z,t)}\mathsf{E}\left[\|g - f\|_{D^{-1}}^2\right] + \sum_{i=1}^N \partial_{\Sigma(z,t)}\mathsf{E}\left[\ln p(x_i\mid y_i)\right]\delta(t - t_i).
$$

We compute $\mathsf{E}\left[\|g - f\|_{D^{-1}}^2\right]$ for linear prior models as

$$
\begin{aligned}
\mathsf{E}\left[\|g - f\|_{D^{-1}}^2\right] &= \mathsf{E}\left[(\bar{A}(z,t)y + \bar{b}(z,t))^\top D^{-1}(\bar{A}(z,t)y + \bar{b}(z,t))\right] \\
&= \sum_z q(z,t)\Big\{\operatorname{tr}\left\{\bar{A}(z,t)^\top D^{-1}\bar{A}(z,t)\Sigma(z,t)\right\} \\
&\qquad + (\bar{A}(z,t)\mu(z,t) + \bar{b}(z,t))^\top D^{-1}(\bar{A}(z,t)\mu(z,t) + \bar{b}(z,t))\Big\}.
\end{aligned}
$$

Using the Gaussian observation likelihoods we arrive at

$$
\begin{aligned}
\partial_{\mu(z,t)}\ell_\mathbb{Q} = & -q_Z(z,t)\left(\bar{A}(z,t)^\top D^{-1}\bar{A}(z,t)\mu(z,t) + \bar{A}(z,t)^\top D^{-1}\bar{b}(z,t)\right) \\
& +\sum_{i=1}^N q_Z(z,t_i)\Sigma_{\mathrm{obs}}^{-1}(x_i - \mu(z,t_i))\delta(t - t_i),
\end{aligned}
$$

$$
\partial_{\Sigma(z,t)}\ell_\mathbb{Q} = -\frac{1}{2}q_Z(z,t)\bar{A}^\top(z,t)D^{-1}\bar{A}(z,t) - \sum_{i=1}^N q(z,t_i)\frac{1}{2}\Sigma_{\mathrm{obs}}^{-1}\delta(t - t_i).
$$

The solutions to these impulsive differential equations are found as for the Lagrange multiplier $\nu(z,t)$, with reset conditions

$$\lambda(z,t_i) = q_Z(z,t_i)\Sigma_{\text{obs}}^{-1}(x_i - \mu(z,t_i)) + \lambda(z,t_i^-)$$

$$\Psi(z,t_i) = -q_Z(z,t_i)\frac{1}{2}\Sigma_{\text{obs}}^{-1} + \Psi(z,t_i^-),$$

with $\lambda(z,t_i^-) := \lim_{h\searrow 0}\lambda(z,t_i - h)$ and $\Psi(z,t_i^-) := \lim_{h\searrow 0}\Psi(z,t_i - h)$.

Note that the Lagrange multiplier equations (44) scale as $\mathcal{O}\left(n^2|\mathcal{Z}|\right)$ (as there have to be carried out $|\mathcal{Z}|$ matrix-vector multiplications of size $n$) and $\mathcal{O}\left(\frac{n(n+1)}{2}|\mathcal{Z}|\right)$ (as due to the symmetry of $\Psi$, only the upper diagonal has to be propagated). The same holds for the forward equations Eq. (15), while the master equation Eq. (4) scales quadratically in $|\mathcal{Z}|$.

### A.3.4 The Optimal Variational Parameters

To optimize the variational parameters, we employ a heuristic back-tracking line search algorithm, as is standard in the field [55]: we choose as step size $\kappa_i = \gamma^i$, where the exponential decay factor is chosen as $\gamma = 0.5$. We therefore update the current parameter $u(t) \in \{A, b, \tilde{\Lambda}\}$ as

$$u_{\text{new}}(t) = u(t) + \kappa_i \cdot \partial_{u(t)}\ell.$$

If $\mathcal{L}[u_{\text{new}}(t)] \geq \mathcal{L}[u(t)]$, we accept the update. Otherwise, we iterate and re-compute using the new step size $\kappa_{i+1}$.

**Gradients for $A(z,t), b(z,t), \tilde{\Lambda}$** We provide here the explicit gradients with respect to the variational parameters. We have

$$
\begin{aligned}
\partial_{A(z,t)}L &= -\frac{1}{2}\partial_{A(z,t)}\,\mathsf{E}\left[\|g - f\|_{D^{-1}}^2\right] - \lambda^\top(z,t)\mu(z,t) - 2\Psi(z,t)\Sigma(z,t)\\
&= -q_Z(z,t)D^{-1}\left(\bar{A}(z,t)(\mu(z,t)\mu^\top(z,t) + \Sigma(z,t)) + \bar{b}(z,t)\mu^\top(z,t)\right)\\
&\quad - \lambda^\top(z,t)\mu(z,t) - 2\Psi(z,t)\Sigma(z,t).
\end{aligned}
\tag{45}
$$

Similarly, we find

$$
\begin{aligned}
\partial_{b(z,t)}L &= -\partial_{b_q(z,t)}\frac{1}{2}\,\mathsf{E}\left[\|g - f\|_{D^{-1}}^2\right] - \lambda(z,t)\\
&= -q_Z(z,t)D^{-1}\left(\bar{A}(z,t)\mu(z,t) + \bar{b}(z,t)\right) - \lambda(z,t).
\end{aligned}
\tag{46}
$$

Finally,

$$
\begin{aligned}
\partial_{\tilde{\Lambda}_{zz'}(t)}L &= -\partial_{\tilde{\Lambda}_{zz'}(t)}\left[\sum_{z'\in\mathcal{Z}\backslash z}\left\{\tilde{\Lambda}(z,z',t)\ln\frac{\tilde{\Lambda}(z,z',t)}{\Lambda(z,z',t)}\right\} - (\tilde{\Lambda}(z,t) - \Lambda(z,t))\right]\\
&\quad + \nu(z,t)q_Z(z,t) - \nu(z',t)q_Z(z,t)\\
&= q_Z(z,t)\left(-\ln\frac{\tilde{\Lambda}_{zz'}(t)}{\Lambda_{zz'}} + \nu(z,t) - \nu(z',t)\right).
\end{aligned}
\tag{47}
$$

**Variational Initial Conditions** The gradients with respect to the initial conditions result from Pontryagin's maximum principle [49, 55] as

$$
\begin{aligned}
\partial_{\mu(z,0)}L &= \partial_{\mu(z,0)}\,\text{KL}\left(\mathbb{Q}_{Y,Z}^0\|\mathbb{P}_{Y,Z}^0\right) + \lambda(z,0) = 0,\\
\partial_{\Sigma(z,0)}L &= \partial_{\Sigma(z,0)}\,\text{KL}\left(\mathbb{Q}_{Y,Z}^0\|\mathbb{P}_{Y,Z}^0\right) + \Psi(z,0) = 0,\\
\partial_{q_Z(z,0)}L &= \partial_{q_Z(z,0)}\,\text{KL}\left(\mathbb{Q}_{Y,Z}^0\|\mathbb{P}_{Y,Z}^0\right) + \nu(z,0) = 0.
\end{aligned}
\tag{48}
$$

While in principle, one could use these expressions to find closed-form solutions for the initial parameters, this is not possible in practice for $\Sigma(z,0)$ and $q_Z(z,0)$. Also, resetting the parameters

may cause numerical instabilities in the forward-backward sweeping algorithm for the constraints and the Lagrange multipliers, cf. Section 3. We hence utilize the same gradient ascent update scheme as above.

We assume a Gaussian prior initial distribution, i.e. $p(y, 0 \mid z) = \mathcal{N}(y \mid \mu_p^0(z), \Sigma_p^0(z))$, and for the presented variational ansatz we have an initial variational distribution $q(y, 0 \mid z) = \mathcal{N}(y \mid \mu(z, 0), \Sigma(z, 0))$, which is also Gaussian. This yields

$$
\begin{aligned}
\mathrm{KL}\left(\mathbb{Q}_{Y|Z}^0 \parallel \mathbb{P}_{Y|Z}^0\right) = {} & \mathrm{KL}\left(\mathcal{N}(y \mid \mu(z, 0), \Sigma(z, 0)) \| \mathcal{N}(y \mid \mu_p^0(z), \Sigma_p^0(z))\right) \\
= {} & \frac{1}{2}\left\{\ln \frac{|\Sigma_p^0(z)|}{|\Sigma(z, 0)|} + \mathrm{tr}\left(\Sigma_p^0(z)^{-1}\Sigma(z, 0)\right)\right. \\
& \left. + \left(\mu_p^0(z) - \mu(z, 0)\right)\Sigma_p^0(z)^{-1}\left(\mu_p^0(z) - \mu(z, 0)\right)^\top + n\right\}.
\end{aligned}
$$

We readily compute

$$
\begin{aligned}
\partial_{\mu(z,0)} \mathrm{KL}\left(\mathbb{Q}_{Y,Z}^0 \| \mathbb{P}_{Y,Z}^0\right) &= q_Z(z, 0)\Sigma_p^0(z)^{-1}(\mu(z, 0) - \mu_p^0(z)), \\
\partial_{q_Z(z,0)} \mathrm{KL}\left(\mathbb{Q}_{Y,Z}^0 \| \mathbb{P}_{Y,Z}^0\right) &= \partial_{q_Z(z,0)}\left\{\mathrm{KL}\left(\mathbb{Q}_Z^0 \| \mathbb{P}_Z^0\right)\right\} + \mathrm{KL}\left(\mathbb{Q}_{Y|Z}^0 \| \mathbb{P}_{Y|Z}^0\right).
\end{aligned}
\tag{49}
$$

For the covariance matrix $\Sigma(z, 0)$ and the initial distribution $q_Z(z, 0)$, we require additional constraints. The covariance $\Sigma(z, 0)$ needs to be positive semi-definite, which can be enforced by a reparameterization as $\Sigma(z, 0) = CC^\top$. We calculate the gradient with respect to the objective

$$
\mathcal{L}_{\mathrm{aug}}(C) = q(z, 0)\left\{\mathrm{KL}\left(\mathcal{N}(y \mid \mu(z, 0), CC^\top) \| \mathcal{N}(y \mid \mu_p^0(z), \Sigma_p^0(z))\right)\right\} + \mathrm{tr}\left(\Psi(z, 0)^\top CC^\top\right)
$$

and the PyTorch package for automatic differentiation and optimization [A2].

The initial distribution $q_Z(z, t)$ needs to fulfil $\sum_z q_Z(z, 0) = 1$, so we optimize an augmented cost function

$$
\begin{aligned}
\mathcal{L}_{\mathrm{aug}}(\mathbb{Q}_Z^0, \xi) = {} & \mathrm{KL}\left(\mathbb{Q}_Z^0 \| \mathbb{P}_Z^0\right) + \sum_z q_Z(z, 0)\,\mathrm{KL}\left(\mathbb{Q}_{Y|Z}^0 \| \mathbb{P}_{Y|Z}^0\right) \\
& + \sum_z \nu(z, 0)q_Z(z, 0) + \xi(1 - \sum_z q_Z(z, 0)),
\end{aligned}
$$

where $\xi(z)$ are Lagrange multipliers. We can again eliminate the constraints by enforcing a reparameterization [A3] as $q_Z(z, 0) = q_z$ for $z \in \{1, \ldots, k-1\}$, with $k = |\mathcal{Z}|$ and $q_Z(k, 0) = 1 - \sum_{z=1}^{k-1} q_z$, which yields the unconstrained problem

$$
\begin{aligned}
\mathcal{L}_{\mathrm{aug}}(q_1, \ldots, q_{k-1}) = {} & \sum_{z=1}^{k-1} q_z \ln \frac{q_z}{p(z, 0)} + (1 - \sum_{z=1}^{k-1} q_z) \ln \frac{1 - \sum_{z=1}^{k-1} q_z}{p(k, 0)} \\
& + \sum_{z=1}^{k-1} q_z\,\mathrm{KL}\left(\mathbb{Q}_{Y|Z=z}^0 \| \mathbb{P}_{Y|Z=z}^0\right) + (1 - \sum_{z=1}^{k-1} q_z)\,\mathrm{KL}\left(\mathbb{Q}_{Y|Z=k}^0 \| \mathbb{P}_{Y|Z=k}^0\right) \\
& + \sum_{z=1}^{k-1} \nu(z, 0)q_z + \nu(k, 0)(1 - \sum_{z=1}^{k-1} q_z).
\end{aligned}
$$

We find

$$
\begin{aligned}
\partial_{q_z} \mathcal{L}_{\mathrm{aug}} = {} & \ln \frac{q_z}{p(z, 0)} + 1 - \ln \frac{1 - \sum_{z=1}^{k-1} q_z}{p(k, 0)} - 1 \\
& + \mathrm{KL}\left(\mathbb{Q}_{Y|Z=z}^0 \| \mathbb{P}_{Y|Z=z}^0\right) - \mathrm{KL}\left(\mathbb{Q}_{Y|Z=k}^0 \| \mathbb{P}_{Y|Z=k}^0\right) + \nu(z, 0) - \nu(k, 0) \\
= {} & \ln \frac{q_z p(k, 0)}{(1 - \sum_{z=1}^{k-1} q_z)p(z, 0)} \\
& + \mathrm{KL}\left(\mathbb{Q}_{Y|Z=z}^0 \| \mathbb{P}_{Y|Z=z}^0\right) - \mathrm{KL}\left(\mathbb{Q}_{Y|Z=k}^0 \| \mathbb{P}_{Y|Z=k}^0\right) + \nu(z, 0) - \nu(k, 0).
\end{aligned}
\tag{50}
$$

| | |
|---|---|
| $A_p$ | prior slope |
| $b_p$ | prior intercept |
| $\Lambda_{zz'}$ | prior transitions rates |
| $\Sigma_{\text{obs}}$ | observation covariance |
| $D$ | Dispersion matrix |
| $\mu_p(z,0), \Sigma_p(z,0), p(z,0)$ | Prior initial conditions |

Table 1: Model parameters learned via VEM

### A.3.5 Optimizing the Prior Parameters

The parameters of the original process $p$, see Table 1, can be learned straightforwardly by optimizing the full Lagrangian Eq. (16) with respect to them.

**Prior MJP transition rates** With the usual shorthand $\Lambda_{zz'}(t) = \Lambda(z,z',t)$, we compute the prior transition rates (which we in all cases assume to be time-homogeneous, $\Lambda_{zz'}(t) = \Lambda_{zz'}$):

$$\frac{\partial L}{\partial \Lambda_{ij}} = -\frac{\partial}{\partial \Lambda_{ij}} \int_0^T \sum_z q_Z(z,t) \sum_{z' \in \mathcal{Z} \backslash z} \left\{ \tilde{\Lambda}_{zz'}(t) \ln \frac{\tilde{\Lambda}_{zz'}(t)}{\Lambda_{zz'}} \right\} - (\tilde{\Lambda}(z,t) - \Lambda(z)) \mathrm{d}t \tag{51}$$

$$= \frac{\partial}{\partial \Lambda_{ij}} \int_0^T \sum_z q_Z(z,t) \left[ \sum_{z' \in \mathcal{Z} \backslash z} \left\{ \tilde{\Lambda}_{zz'}(t) \ln \Lambda_{zz'} - \Lambda_{zz''} \right\} \right] \mathrm{d}t \tag{52}$$

$$= \frac{1}{\Lambda_{ij}} \int_0^T q_Z(i,t) \tilde{\Lambda}_{ij}(t) \mathrm{d}t - \int_0^T q_Z(i,t) \mathrm{d}t. \tag{53}$$

Setting this to zero yields

$$\Lambda_{ij} = \frac{\int_0^T q_Z(i,t) \tilde{\Lambda}_{ij}(t) \mathrm{d}t}{\int_0^T q_Z(i,t) \mathrm{d}t}. \tag{54}$$

**Observation covariance** To determine the observation covariance, we compute

$$\frac{\partial L}{\partial \Sigma_{\text{obs}}^{-1}} = -\frac{\partial}{\partial \Sigma_{\text{obs}}^{-1}} \mathsf{E} \left[ \sum_i \ln p(x_i \mid y_i) \right] \tag{55}$$

$$= -\frac{N}{2} \Sigma_{\text{obs}} + \sum_{i=1}^N \frac{1}{2} \sum_{z \in \mathcal{Z}} q_Z(z,t_i) \left[ (x_i - \mu(z,t_i))(x_i - \mu(z,t_i))^\top + \Sigma(z,t_i) \right], \tag{56}$$

yielding

$$\Sigma_{\text{obs}} = \frac{1}{N} \sum_{i=1}^N \sum_{z \in \mathcal{Z}} q_Z(z,t_i) \left[ (x_i - \mu(z,t_i))(x_i - \mu(z,t_i))^\top + \Sigma(z,t_i) \right]. \tag{57}$$

**Dispersion** We update the dispersion in the same way as the variational parameters (c.f. Appendix A.3.4) and provide the gradient with respect to the dispersion $D$; note that the more general mode-dependent dispersion $D(z)$ are found in the same way by omitting the summation over $z$.

$$\partial_D L = \partial_D \frac{1}{2} \int_0^T \mathsf{E} \left[ \|g - f\|_{D^{-1}}^2 \right] \mathrm{d}t + \partial_D \sum_z \int_0^T \mathrm{tr} \left\{ \Psi^\top(z,t) D \right\} \mathrm{d}t$$

$$= \frac{1}{2} \int_0^T \partial_D \mathsf{E} \left[ \|g - f\|_{D^{-1}}^2 \right] \mathrm{d}t + \int_0^T \sum_{z \in \mathcal{Z}} \Psi(z,t) \mathrm{d}t$$

$$= \frac{1}{2} - D^{-\top} \left( \int_0^T \sum_{z \in \mathcal{Z}} q_Z(z,t) \mathsf{E} \left[ (\bar{A}(z,t)y + \bar{b})(\bar{A}(z,t)y + \bar{b})^\top | z \right] \right) D^{-\top} \tag{58}$$

$$+ \int_0^T \sum_{z \in \mathcal{Z}} \Psi(z,t) \mathrm{d}t.$$

Note that the prior initial conditions $\mu_p^0(z), \Sigma_p^0(z), p(z,0)$ trivially minimize their KL divergence to the variational initial conditions by equality.

**Prior drift parameters** Also the prior parameters $A_p, b_p$ are learned utilizing VEM, see Appendix A.3.4. The gradients are found as

$$\partial_{A_p(z)} L = \partial_{A_p(z)} \frac{1}{2} \int_0^T \mathsf{E}\left[\|g - f\|_{D^{-1}}^2\right] \mathrm{d}t \tag{59}$$

$$= \frac{1}{2} \int_0^T q(z,t) \partial_{A_p(z)} \left\{ \mathrm{tr}\left\{ \bar{A}(z,t)^\top D^{-1} \bar{A}(z,t) \Sigma(z,t) \right\} \right.$$

$$= -\int_0^T q_Z(z,t) \left( D^{-1} \bar{A}(z,t) \left( \Sigma(z,t) + \mu(z,t)\mu^\top(z,t) \right) + D^{-1}\bar{b}(z,t)\mu^\top(z,t) \right) \mathrm{d}t,$$

$$\partial_{b_p(z)} L = \partial_{b_p(z)} \frac{1}{2} \int_0^T \mathsf{E}\left[\|g - f\|_{D^{-1}}^2\right] \mathrm{d}t \tag{60}$$

$$= \int_0^T q_Z(z,t) D^{-1}(\bar{A}(z,t)\mu(z,t) + \bar{b}(z,t))$$

## B  Experiments

### B.1  Model Validation on Ground-Truth Data

A comprehensive overview over the ground-truth and learned parameters is given in Table 2. We model the dispersion as constant, $D(z,t) = D$ and the underlying prior MJP as time-homogeneous, $\Lambda(z,z',t) = \Lambda(z,z')$. The prior drift function reads $f(y,z,t) = \alpha_z(\beta_z - y)$. We will use this parameterization in the following; to convert between this and the hitherto used $f(y,z,t) = A_p(z)y + b_p(z)$, use

$$\begin{aligned} A_p(z) &= -\alpha_z \\ b_p(z) &= \alpha_z \beta_z. \end{aligned} \tag{61}$$

We draw the observation times from a Poisson point process with intensity $\frac{1}{\lambda} = 0.35$, meaning that the average inter-observation interval is $0.35$.

We initialize our model empirically by running a k-means algorithm with $k = 2$ [10] on the observed data. Note that we utilize this procedure for all experiments. The initial prior means and covariances, $\mu_p(z,0)$ and $\Sigma_p(z,0)$, are then set as the cluster means and intra-cluster covariances. The prior intercept $b_p(z)$ is set in the same way. The initial observation covariance $\Sigma_{\mathrm{obs}}$ as well as the dispersion $D$ are both set as the average of the intra-cluster covariances. We can not easily initialize the prior rates $\Lambda$ and the prior slope $A_p(z)$ empirically. We set

$$\Lambda = \begin{pmatrix} -1 & 1 \\ 1 & -1 \end{pmatrix}$$

and $A_p(z) = \{-1, -1\}$. The initial prior $p(z,0)$ is initialized uniformly. The corresponding variational quantities such as $A(z,t)$, are initialized as constant functions on the initial value, e.g.,

$$A(z,t) = A_p(z) \, \forall t.$$

We generate samples from the variational posterior to demonstrate the quality of the latent trajectory reconstruction, see Fig. 6. To sample from the posterior MJP with time-inhomogeneous rates $\tilde{\Lambda}$, we utilize the thinning algorithm [A4].

We show the trajectory of the ELBO over VI iteration in Fig. 7. Furthermore, we demonstrate in Fig. 7 how the framework performs if the requirement of a separation of time-scales is not met. Here, the relaxation of the continuous process is slow compared to the switching process; in other words, a switch in the discrete process is not directly reflected in the continuous state. In this setting, an accurate reconstruction of the latent process is not possible utilizing the variational approximation.

| Parameter | Ground truth value | Learned value |
|---|---|---|
| $\alpha_z$ | $[1.5, 1.5]$ | $[1.04, 0.97]$ |
| $\beta_z$ | $[-1, +1]$ | $[-0.52, 0.70]$ |
| $\Lambda$ | $\begin{pmatrix} -0.2 & 0.2 \\ 0.2 & -0.2 \end{pmatrix}$ | $\begin{pmatrix} -0.68 & 0.68 \\ 0.60 & -0.60 \end{pmatrix}$ |
| $\Sigma_{\mathrm{obs}}$ | $0.1$ | $0.21$ |
| $D$ | $0.25$ | $0.15$ |
| $\mu_p(z, 0)$ | $[-1, +1]$ | $[-0.52, 1.23]$ |
| $\Sigma_p(z, 0)$ | $[0.2, 0.2]$ | $[0.26, 0.03]$ |
| $p(z, 0)$ | $[0, 1]$ | $[0, 1]$ |

Table 2: Ground truth and learned parameters of the 1D, two-mode hybrid process.

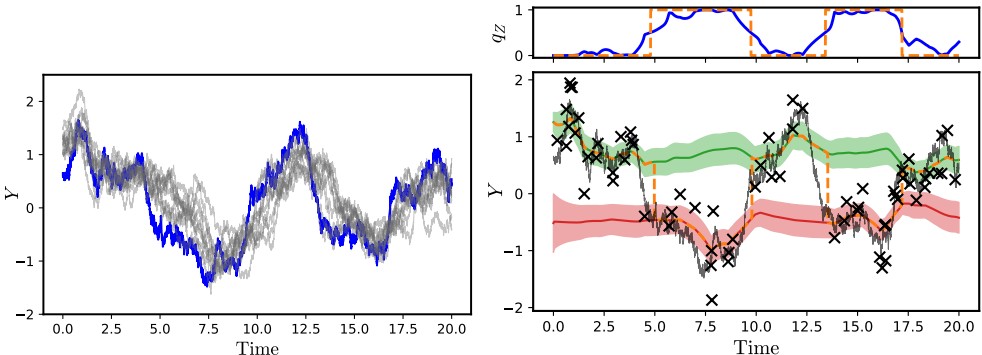

Figure 6: 1D, two-mode hybrid process. Left: posterior samples from the variational distribution (gray) and the latent ground-truth trajectory (blue). Right: Same plot as Fig. 2 A in the main paper, but with both individual modes (red and green) resolved over the complete time span.

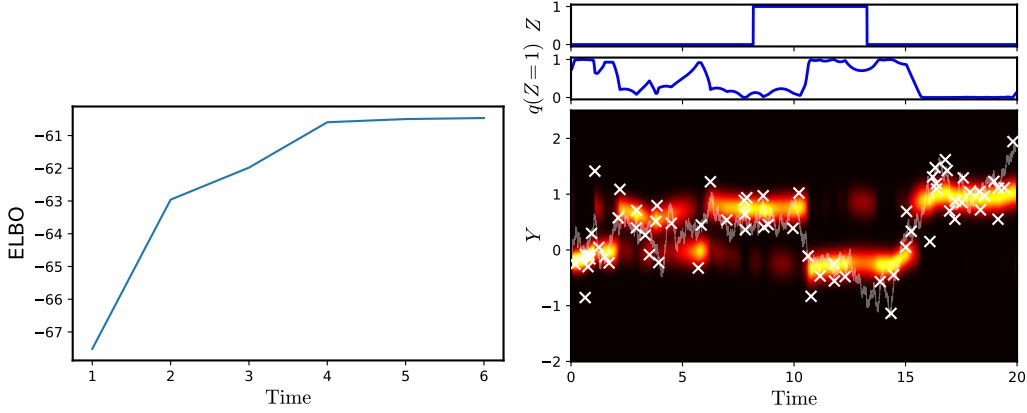

Figure 7: 1D, two-mode hybrid process. Left: value of the ELBO over iterations for the experiment shown in the main paper. Right: Failure of the method when no separation of time scales between the discrete and the continuous process is present.

| Parameter | Learned value |
|---|---|
| $\alpha_z$ | $[1.11, 0.94, 0.99, 0.99]$ |
| $\beta_z$ | $[0.79, -0.27, 0.25, -0.69]$ |
| $\Lambda$ | $\begin{pmatrix} -0.69 & 0.14 & 0.48 & 0.07 \\ 0.07 & -1.39 & 0.45 & 0.87 \\ 1.11 & 0.87 & -2.33 & 0.35 \\ 0.04 & 1.44 & 0.50 & -1.98 \end{pmatrix}$ |
| $D$ | $[0.015, 0.001, 0.003, 0.01]$ |
| $\mu_p(z,0)$ | $[0.99, -0.24, 0.25, -0.69]$ |
| $\Sigma_p(z,0)$ | $[0.002, 0.009, 0.014, 0.025]$ |
| $p(z,0)$ | $[0.92, 0, 0.007, 0.073]$ |

Table 3: Learned parameters of the 1D diffusion in a 4-well potential.

| Parameter | Learned value |
|---|---|
| $\alpha_z$ | $\begin{pmatrix} 0.91 & -0.03 \\ 0.05 & 1.02 \end{pmatrix}, \begin{pmatrix} 1.07 & 0.04 \\ 0.09 & 0.99 \end{pmatrix}, \begin{pmatrix} 1.01 & -0.04 \\ 0.02 & 1.05 \end{pmatrix}$ |
| $\beta_z$ | $\begin{pmatrix} 1.08 \\ 0.28 \end{pmatrix}, \begin{pmatrix} -0.78 \\ -0.17 \end{pmatrix}, \begin{pmatrix} 0.02 \\ 1.10 \end{pmatrix}$ |
| $\Lambda$ | $\begin{pmatrix} -0.75 & 0.29 & 0.46 \\ 0.26 & -0.75 & 0.49 \\ 0.80 & 0.77 & -1.57 \end{pmatrix}$ |
| $D$ | $\begin{pmatrix} 0.048 & -0.003 \\ -0.003 & 0.077 \end{pmatrix}, \begin{pmatrix} 0.088 & -0.035 \\ -0.035 & 0.092 \end{pmatrix}, \begin{pmatrix} 0.077 & -0.004 \\ -0.004 & 0.040 \end{pmatrix}$ |
| $\mu_p(z,0)$ | $\begin{pmatrix} 1.18 \\ 0.11 \end{pmatrix}, \begin{pmatrix} -0.41 \\ -0.01 \end{pmatrix}, \begin{pmatrix} -0.54 \\ 0.03 \end{pmatrix}$ |
| $\Sigma_p(z,0)$ | $\begin{pmatrix} 0.263 & 0.058 \\ 0.058 & 0.269 \end{pmatrix}, \begin{pmatrix} 0.029 & 0.002 \\ 0.002 & 0.030 \end{pmatrix}, \begin{pmatrix} 0.090 & 0.002 \\ 0.002 & 0.090 \end{pmatrix}$ |
| $p(z,0)$ | $[0.021, 0.173, 0.81]$ |

Table 4: Learned parameters of the 2D diffusion in a 3-well potential.

### B.2 Diffusions in Multi-Well Potentials

The one-dimensional 4-well potential is given as

$$V(y) = 4\left( y^8 + 3e^{-80y^2} + 2.5e^{-80(y-0.5)^2} + 2.5e^{-80(y+0.5)^2} \right), \tag{62}$$

the two-dimensional 3-well potential reads

$$V(y_1, y_2) = 3e^{-y_1^2 - (y_2 - \frac{1}{3})^2} - 3e^{-y_1^2 - (y_2 - \frac{5}{3})^2} - 5e^{-(y_1 - 1)^2 - y_2^2}$$
$$- 5e^{-(y_1 + 1)^2 - y_2^2} + 0.2y_1^4 + 0.2\left( y_2 - \frac{1}{3} \right)^4. \tag{63}$$

In the 1D example, we fix the observation covariance as $\Sigma_{\text{obs}} = 0.0225$. We provide the list of all learned parameters in Table 3. In the 2D example, we fix $\Sigma_{\text{obs}} = \begin{pmatrix} 0.2 & 0 \\ 0 & 0.2 \end{pmatrix}$; the exhaustive list of parameters is given in Table 4. The initialization is done as in Appendix B.1.

### B.3 Switching Ion Channel Data

The experimental data have been obtained using a voltage of $140\,\text{mV}$ and a sampling frequency of $5\,\text{kHz}$ over a measurement period of $1\,\text{s}$. We fix the observation noise to $\sqrt{\Sigma_{\text{obs}}} = \sigma_{\text{obs}} = 0.25\,\text{fA}$. The full list of parameters is given in Table 5. Initialization is done as in Appendix B.1, but with $\Lambda_{z,z'} = 100$ for $z \neq z'$.

| Parameter | Learned value |
|---|---|
| $\alpha_z$ | $[1,1,1]$ |
| $\beta_z$ | $[6.2 \cdot 10^{-12}, 4.4 \cdot 10^{-14}, 3.8 \cdot 10^{-12}]$ |
| $\Lambda$ | $\begin{pmatrix} -13.01 & 2.86 & 10.15 \\ 11.58 & -30.16 & 18.58 \\ 44.33 & 23.35 & -67.68 \end{pmatrix}$ |
| $D$ | $[0.015, 0.001, 0.003, 0.01]$ |
| $\mu_p(z,0)$ | $[6.2 \cdot 10^{-12}, 4.4 \cdot 10^{-14}, 3.8 \cdot 10^{-12}]$ |
| $\sqrt{\Sigma_p(z,0)}$ | $[2.9 \cdot 10^{-13}, 3.7 \cdot 10^{-13}, 6.1 \cdot 10^{-13}]$ |
| $p(z,0)$ | $[1,0,0]$ |

Table 5: Learned parameters for ion channel data.

| Parameter | Ground truth value | Learned value |
|---|---|---|
| $\alpha_z$ | $\begin{pmatrix} 0.6 & -1.4 \\ 2.6 & 0.6 \end{pmatrix}, \begin{pmatrix} -0.1 & 1.4 \\ -2.6 & 0.6 \end{pmatrix}$ | $\begin{pmatrix} 0.41 & -0.83 \\ 1.04 & -0.01 \end{pmatrix}, \begin{pmatrix} -0.45 & 1.49 \\ -1.95 & 0.38 \end{pmatrix}$ |
| $\beta_z$ | $\begin{pmatrix} -5 \\ 0 \end{pmatrix}, \begin{pmatrix} 5 \\ 0 \end{pmatrix}$ | $\begin{pmatrix} -6.00 \\ -1.79 \end{pmatrix}, \begin{pmatrix} 5.85 \\ -2.20 \end{pmatrix}$ |
| $\Lambda$ | $\begin{pmatrix} -0.3 & 0.3 \\ 0.3, -0.3 \end{pmatrix}$ | $\begin{pmatrix} -0.17 & 0.17 \\ 0.23, -0.23 \end{pmatrix}$ |
| $D$ | $\begin{pmatrix} 0.49 & 0 \\ 0 & 0.49 \end{pmatrix}$ | $\begin{pmatrix} 0.87 & 0.06 \\ 0.06 & 1.46 \end{pmatrix}$ |
| $\mu_p(z,0)$ | $\begin{pmatrix} -5 \\ 0 \end{pmatrix}, \begin{pmatrix} 5 \\ 0 \end{pmatrix}$ | $\begin{pmatrix} -5.38 \\ -0.42 \end{pmatrix}, \begin{pmatrix} 5.67 \\ 1.40 \end{pmatrix}$ |
| $\Sigma_p(z,0)$ | $\begin{pmatrix} 0.49 & 0 \\ 0 & 0.49 \end{pmatrix}, \begin{pmatrix} 0.49 & 0 \\ 0 & 0.49 \end{pmatrix}$ | $\begin{pmatrix} 6.63 & 1.24 \\ 1.24 & 16.00 \end{pmatrix}, \begin{pmatrix} 0.001 & 0.0003 \\ 0.0003 & 0.001 \end{pmatrix}$ |
| $p(z,0)$ | $[0,1]$ | $[0,1]$ |

Table 6: Ground-truth and learned parameters of the complex structured 2D switching diffusion.

## B.4 Learning Complex Latent Continuous Dynamics

Here, observations are drawn from a Poisson point process as in Appendix B.1 with intensity $\lambda = 14$. The ground-truth and learned parameters are summarized in Table 6. Initialization is done as in Appendix B.1.