# OpenReview forum: "Variational Inference for Continuous-Time Switching Dynamical Systems"
_NeurIPS.cc/2021/Conference — NeurIPS 2021 Spotlight_

### Official Review · Reviewer_QSy5 · 2021-07-14

**Rating:** 8
**Confidence:** 4

**Summary:**

In this paper, the authors propose a variational inference framework for continuous time hybrid process models. The particular assumption on the true posterior allows for a mixture of GPs as an approximation to it. The proposed framework is tested on various benchmark tasks including real-world biological data and the results demonstrate its ability to faithfully reconstruct complex latent dynamics and to learn the unknown system parameters, in particular in applications to meta-stable systems. It extends the toolbox for inference will be of great utility for the analysis and control of such hybrid systems.

**Limitations And Societal Impact:**

The authors do not talk much about limitations or societal impact.


**Main Review:**

Originality: The proposed method is new for continuous-time SLDS. The assumed variational posterior enables inference via optimization. The relevant previous contributions were clearly discussed and cited.

Quality: The submission is technically sound. The methodology is rigorously derived and empirically tested with various synthetic and real data. The work in this manuscript is a complete piece of work. The weaknesses of the work was not substantially evaluated, especially for non-meta-stable systems like continuous attractors. The experiments shown in the manuscript do not include a case that the number of state $Z$ is mismatched. Since the optimization of ELBO is generally a non-convex problem, the authors better talk about the convergence, e.g. empirically convergence reached and converging to the same or different optima with different initial values, and if the optimas are equivalent.

Clarity: Overall, the manuscript was well organized and written clearly. See itemized comments.
- Line 202: Where are $A_p$ and $b_p$ defined? What does the subscript $p$ represent, prior? Are they assumed constant as $A$ and $b$ latter on.
- Fig2 B: I suppose the curve attached to the right is the wells, but it is not described in the caption.

Significance: This work extends SLDS to continuous-time regime and proposes easy approximate inference via optimization. SLDS is well known in e.g. the field neuroscience for modeling meta-stable dynamics. This work would have high impact in such fields and be of great utility for the analysis and control of such hybrid systems.

**Time Spent Reviewing:**

8

---

> ### Author Response · Authors · 2021-08-09
> **Addressing comments and questions**
>
> Thank you for your constructive and thoughtful comments.
> We introduce our approximation specifically for meta-stable systems; an in-depth analysis of its performance in settings that are outside of this ‘domain of validity’ would be out of scope for the present work, also in view of space constraints. To give a better intuition, we will add a version of experiment 1 to the supplement where the relaxation to the continuous set-points is much slower than in the paper version (no separation of time scales any more) and, consequentially, the framework fails to recover the ground truth. We do agree that in order to generalize the framework, this issue should be thoroughly analyzed in future works. This should also include the problem of the number of modes, which might in many real-world situations not be known in advance: we plan on investigating this in a follow-up study (see also discussion with reviewer naZJ).
>
> We assessed convergence empirically by tracking the value of the ELBO over iterations; if its relative change fell below a value of 1e-3, we took this as converged. Note that the problem of different local optima reached from different initial values can be alleviated by using multi-start methods. We will extend Sect. 3.3 to include this information in the camera-ready version and also include an example of the ELBO over iterations in the supplement. We note that as this is not the focus of our work, we opted for a basic and simple-to-implement scheme (gradient ascent with back-tracking line search), which can be readily replaced on demand.
>
> We apologize for not making clear that the drift function in line 202 defines A and b; the subscript indeed is short for ‘prior’, which we will make explicit in the final version of the manuscript. We will also clarify that, while in general, both may be time-dependent, we constraint them to be constant in all experiments.
>
> We do mention that the curve in Fig. 2 B represents the wells: “Figure 2: […] B: Diffusion in a 1D four-well potential (shown right). […]”. We acknowledge that it may be unclear that “shown right” refers to “four-well potential” only, rather than the complete sentence; we will make this clear in the final paper version.

---

> > ### Comment · Reviewer_QSy5 · 2021-08-16
> > **My recommendation remains the same**
> >
> > Thank you for the response. My concerns are addressed. My recommendation remains the same.

---

### Official Review · Reviewer_d46C · 2021-07-16

**Rating:** 7
**Confidence:** 2

**Summary:**

The paper proposes a variational inference for continuous-time switching dynamical systems, whose inference is believed to be lacking in need of biology and engineering applications. The form of the variational distributions is constrained by a master equation for the jump process and a Fokker-Planck equation for the diffusion process. These constraints are encoded in the Lagrangian. Both model parameters and variational distribution are inferred by a variational EM algorithm. The validity of the model is demonstrated with low-dimensional experiments.

**Limitations And Societal Impact:**

I didn't find the limitation statements, although the checklist indicates there is one in a quite large page range. It would be better to describe the computation aspects and whether it fits modern high-dimensional applications.

**Main Review:**

The derivation is very fundamental, and the statement is mathematically rigorous. It starts from the Markov process formulation and arrives at the inference algorithm for switching dynamical systems. My main concern is that whether such lower-level formulation is required for applications. The following are specific concerning points and please correct me if my understanding is wrong.
- By the formulation, since continuous variables in both variational processes and the prior processes have linear drift, can we have explicit mean and variance representations for the diffusion process, e.g., Ornstein-Uhlenbeck process has an explicit time-dependent mean and variance? If this is possible, then we can work with a parametric Gaussian as the diffusion process and are free from the constraints.
- The experiments are conducted in low-dimensional space, does it work in a high-dimensional setup? There are some works using diffusion processes for deep models or high-dimensional data.
[Kurle, Richard, et al. "Continual learning with bayesian neural networks for non-stationary data." International Conference on Learning Representations. 2019.]
[Bamler, Robert, and Stephan Mandt. "Dynamic word embeddings." International conference on Machine learning. PMLR, 2017.]


**Time Spent Reviewing:**

3

---

> ### Author Response · Authors · 2021-08-09
> **Addressing questions and comments**
>
> Thank you for your thoughtful feedback.
> We believe that the introduction of a new inference framework requires a mathematically rigorous, ‘low-level’ formulation, which was the main goal of our work. Furthermore, since application to any specific research or engineering area may require one to adapt, e.g., the constraints to the problem at hand, a sound understanding of the underlying machinery is also necessary when one just wants to apply the method.
>
> We do indeed represent the diffusion process via time-dependent mean and covariance functions; in other words, we utilize exactly the Gaussian process approximation.
>
> The transfer to high-dimensional settings is a non-trivial problem already for conventional diffusion processes and requires additional approximations. For instance, to be able to recover complex dynamics as in 4.4 in higher dimensions, one cannot impose a diagonality constraint on the SDE dynamics A(z, t) as done in Vrettas, Opper, Cornford, Phys Rev E 2015. A thorough investigation is out of scope for the present study and hence left for future work. See also the discussion with reviewer naZJ.

---

> > ### Comment · Reviewer_d46C · 2021-08-16
> > **Need further clarifications**
> >
> > Thanks for addressing my concerns. I acknowledge that the contribution is a solid inference framework for a continuous-time switching dynamical system. For the math-rigorous work of this type, my understanding is that by solving a low-level problem, the work will usually provide insights, intuitions, or implications for other applications. I failed to see this and I will appreciate it that the authors could provide further clarification.
> >
> > I want to clarify one of my reviews. Sorry for my previous wording and the confusion it causes: when I said "explicit" representations, I really meant *analytic* representations; otherwise, it is trivial to say so. With this in mind, the experiments in the paper may (probably) also be tackled with existing switching linear dynamical system solvers, e.g., a Gaussian sum smoother, with a temporal notion introduced for the latent process. Without addressing this or providing comparisons, the experiments don't distinguish your solver from others.
> >
> > That said, I'm willing to increase my score if my above concerns can be addressed. Thanks!

---

> > > ### Author Response · Authors · 2021-08-18
> > > **Further clarifications**
> > >
> > > Thank you for your remarks, which we will clarify in order.
> > >
> > > We apologize if we did not make clear enough what our framework achieves and what its implications are. An important insight following from our work is that the hybrid process model yields sensible, well interpretable results for non-hybrid systems. We demonstrated and discussed this via the experiments in Sect. 4.2 and 4.3. This shows that our framework enables accurate inference (i) in time-continuous models for such processes, and (ii) within an explicit, unified account for both latent continuous and discrete dynamics. Until now, only discrete-time frameworks are available. Regarding (ii), we note as an example that in Markov state modeling of molecular dynamics data, the different process classes are typically dealt with one-after-the-other in an ad-hoc, multi-step procedure, cf. [48-51], as no unified framework is available. Accordingly, our results imply a wide range of potential application areas that are concerned with continuous state processes exhibiting functionally distinct regimes. In addition to the references given in the introduction, see also, for instance, [R1-R6]. Another direct implication is that there presumably is a large potential for future work and improvements building on our approach, further drawing on developments and established methods in the fields of variational inference and optimal control.
> > > A more technical, but central insight lies in the fact that the PDE (7) can be well approximated using a mixture of GPs for meta-stable systems. As (7) is a complex quantity with mixed continuous-discrete state space, it does not provide much intuition about the described process. This intuition can however be gained via the approximation: for meta-stable systems, the PDE (roughly) describes a continuous process switching between distinct modes. In other words, at any given time point, the process can be clearly assigned to one of the modes.
> > >
> > > As to the question of analytic representations, we note that the ODEs (15) for the Gaussian parameters do not have general, analytic solutions, as (i) their parameters $A(z, t), b(z, t)$ are time-dependent and (ii) they are *impulsive* ODEs [A2], which are reset at the observation time points as detailed in Sect. A.3.3. For general linear ODEs $\dot{\mathbf{x}}(t) = \mathbf{F}(t)\mathbf{x}(t)$ with time-dependent parameters, a solution can be formulated in terms of a transition matrix $\boldsymbol{\Psi}(t, t_0)$, which itself is defined via a set of ODEs. These do however not have an analytic solution in general (for more details, see, e.g., [27]), and can only be represented in terms of an infinite Peano-Baker series [R7]. We note that *even if* an analytic solution of the type $\mu(z, t) \propto \exp\{A\mu(z, t)\}$ were available, a standard way of computing the matrix exponential would in fact be to solve the ODE numerically, see [R8]. In general, while recursive solutions for many filtering and smoothing problems are well established (see, e.g., the famous Kalman filter or the Rauch-Tung-Striebel smoother), closed-form solutions can only be found for the simplest systems (cf. the examples in [27]). Note that regardless of the solution method, our framework does work, as stated in your original review, with parametric Gaussians as the latent diffusion processes. We cannot get rid of constraints in this way, however: the constraint equations (15) precisely prescribe the Gaussian parameters.
> > >
> > > We acknowledge that we did not reference the Gaussian sum smoother as a solution method to switching linear dynamical systems [R9]; thanks for bringing this to our attention. We will include it in the camera-ready version. We note, however, that the Gaussian sum smoother neither yields closed-form, analytic solutions. In fact, while their approach is conceptually different from ours (working in discrete-time, using ad-hoc approximations on the exact posterior rather than a rigorous variational approach), the recursions for the Gaussian means and covariances obtained in [R9] are qualitatively similar to our ODEs (15).
> > >
> > > Lastly, as to a potential comparison to existing methods, we restate that all existing inference frameworks for SLDS only work in a discrete time setting [6-9, R9]. Since the aim of our contribution was to establish a principled framework for continuous time hybrid systems, we think that a comparison to advanced discrete-time SLDS methods is neither necessary nor would it be fair. Our intention is not to outperform discrete methods. Also, our assumed general setting with non-equally spaced observations does not readily transfer to the discrete-time case. We want to emphasize that the long-standing need for continuous-time filtering and smoothing frameworks is substantiated by the fact that the established discrete-time filters and smoothers are all worked out in continuous-time formulations as well (see, e.g., the continuous-time versions of the Kalman filter and the RTS smoother or the Kalman-Bucy filter, [27]).
> > >
> > > We will use the additional page in the camera-ready version to include the above elaborations, which was not possible in the submission due to space constraints.
> > >
> > > * [R1: Deco et al., “Awakening: Predicting external stimulation to force transitions between different brain states”, PNAS 2019]
> > > * [R2: Kringelbach & Deco, “Brain states and transitions: insights from computational neuroscience”, Cell Reports 2020]
> > > * [R3: Taghia et al., “Uncovering hidden brain state dynamics that regulate performance and decision-making during cognition”, Nat Comm 2018]
> > > * [R4: Costa et al., “Adaptive, locally linear models of complex dynamics”, PNAS 2019]
> > > *[R5: Khaled et al., “Regime switching model for financial data: Empirical risk analysis”, Physica A 2015]
> > > *[R6: Duprey & Klaus, “How to predict financial stress? An assessment of Markov switching models”, ECB Working Paper Series May 2017]
> > > *[R7: Baake & Schlägel, “The Peano-Baker series”, Proceedings of the Steklov Institute of Mathematics 2010]
> > > *[R8: Moler, van Loan, “Nineteen dubious ways to compute the exponential of a matrix, twenty-five years later” SIAM Review, 2003]
> > > *[R9: Barber & Mesot, "A novel Gaussian sum smoother for approximate inference in switching linear dynamical systems" NIPS 2007]

---

> > > > ### Comment · Reviewer_d46C · 2021-08-24
> > > > **Thanks for the response**
> > > >
> > > > Thanks for addressing the points! The importance of the continuous-time switching dynamical system is well explained.

---

### Official Review · Reviewer_naZJ · 2021-07-16

**Rating:** 7
**Confidence:** 3

**Summary:**

The manuscript extends the literature of statistical inference in switching linear systems to the continuous domain. Starting from the specifics of the generative model (a discrete valued latent stochastic Markov Jump process modeling the "mode" of the hybrid process and a continuous valued latent diffusion process modeling the "state" of the process and finally an observation model connecting the latent space to discretely sampled observations) and then provides the expressions for equations governing the joint, posterior (or smoothing) and marginal distributions. The authors then show why exact inference is intractable and provide the methodology needed for approximate (structured mean-field) variational inference. The variational family considered is a mixture of GPs (which turns out to be the case because of the structural PDE equations enforced on the variational family to mimic the true posterior distribution) where the time-specific mixture weights follow a master equation and variational factor q(y,t|z) follows a a Fokker-Planck equation. The ELBO cost function is then derived and optimized using forward-backward sweeping algorithm. To optimize the model parameters (transition rate function, parameters of the drift function, and dispersion) a variational EM approach is taken where the parameters are updated iteratively after satisfying structural constraints and updating variational parameters. Experiments are performed on toy examples as well as multi-well potentials and ion channel data.

**Limitations And Societal Impact:**

For limitations, see above.

As the authors mentioned, the work is theoretical and hence there are not trivial societal and ethical impacts subject to discussion.

**Main Review:**

Strengths
=======
The question under investigation is timely, significant, and relevant to NeurIPS community in my opinion.

The background section includes a nice summary of the prerequisites needed for following the ideas throughout the paper.

The introduction provides a nice methodological literature as well as the application of the methods in various fields.

Although the content of the manuscript is fairly complex I found the flow of the writing very sensible and easy to follow.

The amount of mathematical details included in the main text seems appropriate and every details that is mentioned in the supplementary is properly referenced in the main text.

Finally the experiments are diverse and the visualizations are very helpful for digesting the outputs of the model and its inner workings.

Suggestions
==========
Adding some analytical intuitions (not derivations) would help the reader to have a better sense about the equations.

It would be nice to supplement the visualizations with the observations x(t) to show what type of data is being considered.

Weaknesses
==========
The experiments are performed mostly on the cases where the analytical form of the drift function is known. However in real world scenarios it's rarely the case and in many cases the function is approximated using another parametric family. For discrete cases to the best of my knowledge neural networks are used to model the drift function (e.g. https://arxiv.org/abs/1605.08454). It would be useful to investigate if the model can recover dynamical regimes even in the presence of mismatch in the drift function.

Almost all the model validation that is done in the experiments section is qualitative. Even for a low-D system the parameter error (beta) seems relatively large. A more thorough investigation of the error as a function of dimension, number of modes, and separability between different dynamical regimes would help practitioners to apply the method with some prior caution.

Since the field of statistical inference in discrete switching dynamical systems is fairly stablished it would be nice to perform comparisons between the fit given by the continuous model (proposed) and discrete models.

**Time Spent Reviewing:**

6

---

> ### Author Response · Authors · 2021-08-09
> **Addressing comments**
>
> Thank you for your constructive and thoughtful feedback.
> We mostly (except Fig. 4) visualize the observations using (white or colored) crosses, see for instance Fig. 2.
>
> We focused on linear prior models to keep things as simple and concise as possible, since our main goal was to present the hybrid variational framework as such. To us, it indeed appears to be a natural next step to formulate the latent discrete model as a neural SDE (https://arxiv.org/pdf/2001.01328.pdf). For space and complexity reasons, we leave this for future work. For some further details, please also see the discussion with reviewer czKb.
>
> We do agree that a more quantitative evaluation of the (approximation) errors as a function of (state and mode) dimension is needed for a deeper understanding of the model. However, even for moderate (state) dimensionalities, it is state of the art to introduce further approximations to maintain tractability (Vrettas, Opper, Cornford, Phys Rev E 2015). The number of modes is a variable that has (at the moment) to be given in advance; we plan on doing a further study relaxing this constraint. Hence, before quantitative analyses over a considerable dimensionality range can be carried out for the hybrid process, further work is needed on both processes individually and consequentially is out of scope for the present study.
>
> Lastly, we think that a comparison to advanced discrete-time SLDS methods would not be fair; our intention is not to outperform discrete methods, but to establish a principled framework for continuous time hybrid systems. Additionally, our assumed general setting with non-equally spaced observations does not readily transfer to the discrete-time case.

---

### Official Review · Reviewer_czKb · 2021-07-16

**Rating:** 7
**Confidence:** 3

**Summary:**

This manuscript concerns a continuous-time variant of switching state-space models, aka switching stochastic differential equations (SSDEs). Such hybrid models have a Markov jump process $z_t$ that modulates the diffusion process $y_t$. The contribution of this work is a new and unifying mean-field variational framework that leads to scalable approximation. In particular, time evolutions of the approximate posteriors over the switch variables $q(z,t)$ and the conditional $q(y,t|z)$ are computed individually. The final optimization objective is a Lagrangian consisting of the evidence lower bound (ELBO) and implicit-form ordinary differential equation (ODE) constraints describing the above-mentioned time evolutions. As usual, Lagrange multipliers are given by the optimality conditions defined by the Euler-Lagrange equations. The method is demonstrated on simple toy and real-world datasets with irregular observations. It is shown to consistently learn the continuous dynamics and infer accurate approximate posteriors over the switch $z_t$.

**Limitations And Societal Impact:**

All the limitations and potential negative societal impacts are addressed.

**Main Review:**

With the neural ODE breakthrough and succeeding impressive NODE-based methodologies, continuous-time models have gained huge popularity. So, the manuscript studies a theoretically interesting and timely topic. Unlike existing approaches, the real-world applications of this work might be a little limited (perhaps for the time being) but the rigorous theoretical investigation easily outweighs it.

The proposed variational inference framework leads to computationally convenient expressions as shown by the authors, which is often rare in such hierarchical models. Therefore, although the inference framework overlaps with [20] in large parts, I find the contribution significant.

The authors motivate and describe the problem very well. Despite the heavy use of mathematics, I was able to follow the main arguments of the manuscript. So, I could say that it is written as clear as possible. Also, please note that I was not able to follow all the equations in the Appendix line by line but there should be no issue in the ones in the main paper.

Overall, I recommend an accept as the submission is above average in all acceptance criteria. That being said, I am slightly concerned that the entire framework might rely on a wrong assumption. More specifically, is the factorization in eq.(12) correct? Should not one of the t's be on the rhs (to be conditioned to)? I wonder what the authors (and other reviewers) think about this.

Below are more detailed (minor) comments:
- The approximations typically hold when the posterior concentrates on a single value of $z$. What if it does not? When does it tend to hold?
- I wish to see the examples in which the framework starts failing, which would help us understand the limits of the method. Also, comparing the method with baselines would even further highlight its merits.
- It would be much better if the authors can state which ideas are inspired by which papers. This would allow the reader to check reference papers more easily.
- How important is the linear drift choice? How does the model behave with less/more data?
- The derivation in Section3.2 is deeply connected to Pontryagin's maximum principle, which goes uncited. I suggest the authors check out the connections, which would allow the model to draw more attention.
- Why is back-tracking line search needed? In general, how long does the optimization take? How sensitive is the optimization?
- In line 242, do the left hand side expressions resemble Viterbi path? If so, the relationship should be shown.

**Post-rebuttal comment:** Thanks for the rebuttal. All in all, I believe the paper should be accepted. It would be better if the following are addressed in the final version (or any arxiv submission or similar):
- Lack of quantitative comparisons
- References should be improved (I don't very closely follow the "Switching Dynamical Systems" literature but at least the VI framework is heavily inspired from [20], which is not explicitly mentioned in the text).
- I still believe the factorization in eq.(12) is invalid. I'm starting to think this is just a notation issue.

**Time Spent Reviewing:**

6

---

> ### Author Response · Authors · 2021-08-09
> **Addressing questions and comments**
>
> Thank you for your constructive feedback and comments.
> The factorization of the true posterior in eq. (12) (first line) is just an application of the definition of conditional/joint probabilities; the second line is our approximation (hence no equals-sign). The confusion might stem from the fact that in our notation, we do not re-iterate the time argument ‘t’ in the conditioning set (c.f. the definition of the PDF right after eq. (12)). We opt to do so to keep the notation as compact and uncluttered as possible.
>
> We go through your comments in order:
>
> As discussed in Sect. 3.1, the approximation becomes inaccurate whenever the posterior does not concentrate on a single mode. For systems exhibiting meta-stability---i.e., a separation of time scales between the discrete and continuous latent processes---, the posterior is however expected to concentrate, which is the setting we are interested in. To give a better intuition about the limits of the method, we will add a version of experiment 1 to the supplement where the relaxation to the continuous set-points is much slower than in the paper version (hence, no separation of time scales any more) and, consequentially, the framework fails to recover the ground truth. Since no inference methods for hybrid systems are established, it is however unclear what a sensible baseline for our framework would be; state of the art algorithms can only provide parts of what our model achieves, e.g., SDE smoothing (without a discrete process).
>
> We focused on linear prior models to keep things as simple and concise as possible, since our main goal was to present the hybrid variational framework as such. Non-linear prior models would (i) entail an additional approximation error, if the variational process is still chosen linear and (ii) even for such linear variational models, one would encounter analytically intractable expectations (e.g., $\mathbb{E}[\Vert g – f\Vert^2]$, cf. eq. (11)). Such expressions could be evaluated using sampling techniques, resulting in a set of effective SDEs instead of ODEs for the Lagrange multipliers, eq. (15).  While being outside the scope of the present work, we think this is a very interesting topic that is worth being studied in isolation as a form of stochastic variational inference for continuous-time systems.
>
> Note that since we pursued a Bayesian approach, the method will behave as the specified prior in settings with sparse or even no data. We mentioned this explicitly for the exact posterior, but we will clarify that this also holds for the variational model in the camera-ready version.
>
> It is correct that the results of Sect. 3.2 are based on Pontryagin’s maximum principle (PMP); we mention this explicitly only in the supplement and will add this to the main text in the camera-ready version. Note that the PMP is discussed in-depth in the control literature that we cite (e.g., [43, 44, 45]). Problems as discussed in 3.2 are deeply rooted in control: our variational density can be understood as the state of the control system; the variational parameters ($A, b, \tilde{\Lambda}$) are the controls; and eq. (16) is the objective function, which is to be optimized with respect to the controls. We note that the maximum principle in general is only a necessary, but not a sufficient condition for optimality. In particular, the Euler-Lagrange (EL) equations are not a sufficient condition. This elicits the problem that optimization utilizing the EL equations in general only yields local optima instead of the global one. Conceptually, the global optimum can be found by solving the Hamilton-Jacobi-Bellman (HJB) equation (see, e.g., [43]) which is a necessary and sufficient condition for optimality. The HJB equation however is itself a PDE, which then defeats the purpose of our approximation and hence is not elaborated upon.
>
> We utilize back-tracking line search for numerical stability, since it guarantees finding a gradient step-size that will improve the overall ELBO. We opt for this basic, simple-to-implement scheme as parameter optimization is not the focus of our work and a formidable challenge in itself. It is however conceptually straightforward to utilize more sophisticated methods, such as conjugate gradient schemes. In general, optimization is sensitive to initialization. To overcome this issue, one can for instance employ multi-start methods; we will add this information to Sect. 3.3 in the final version. In general, the optimization speed depends on the objective function, the curvature of which determines the convergence of the gradient steps (see, e.g., [A4], “Condition Number”). Empirically, we observed that only a few steps (on the order of 10) are needed until convergence.
>
> We note that in the MAP-paths as we define, the MAP is taken time-point wise.  This is different from the Viterbi path, which maximizes the whole path-distribution. For diffusion processes, this requires optimizing the Onsager-Machlup functional (see, e.g., [27]). A generalization of this methodology to hybrid systems is non-trivial and left for future work.

---

### Decision · Program_Chairs · 2021-09-28

**Decision:**

Accept (Spotlight)

**Comment:**

This paper presents a model for continuous-time temporal datasets. The model takes the form of a Markov jump process that modulates a diffusion process. The work addresses posterior inference by applying a continuous-time variational inference algorithm that combines Gaussian processes for the diffusion process and posterior inference for the Markov jump process.

The reviewers agreed the paper is timely, significant, and relevant for the community, and that it deserves acceptance.

I encourage the authors to address the concerns raised by the reviewers. In particular:
+ Include a more thorough investigation of the parameter error in different regimes (i.e., dimensions, modes, separability of the modes). A discussion of which of these might be the limiting factor from a practical perspective would be useful for the readers.
+ Improve the references [R1-R9], including the ones in the discussion with reviewer d46C and specially the references about the variational inference method, e.g., mention [20] in the text.
+ Clarify the issue with the factorization in Eq. (12). The authors argued that it's just an attempt to make the notation uncluttered, but I agree with reviewer czKb that it creates confusion and it makes it look like the factorization is incorrect. Consider changing the notation accordingly.
+ Address the other changes that were discussed during the review period (e.g., extend Section 3.3, include an example of ELBO vs iterations, clarify the points about the drift function and about Figure 2, etc.).

**Consistency Experiment:**

NeurIPS has a long history of experimentation. In 2014, NeurIPS ran an experiment in which 10% of submissions were reviewed by two independent committees to quantify the randomness in the review process. This year, we repeated a variant of this experiment to see how the quality of the review process has changed over time.  This paper was part of the experiment and was therefore assigned to two committees (consisting of reviewers, an Area Chair, and a Senior Area Chair) that reached independent decisions.  If both committees made the same recommendation, this recommendation was followed. If a single committee recommended acceptance, the paper was accepted (with the exception of a few cases in which the other committee identified what we considered a fatal flaw, e.g., an error in a key result).

This copy’s committee reached the following decision: **Accept (Spotlight)**

The other committee assigned to the paper recommended **Accept (Poster)**.  You can find the other set of reviews, along with any follow up discussion with the authors here:
https://openreview.net/forum?id=ake1XpIrDKN